# A PKB-SPEG signaling nexus links insulin resistance with diabetic cardiomyopathy by regulating calcium homeostasis

Chao Quan[1,7], Qian Du[1,7], Min Li[1], Ruizhen Wang[1], Qian Ouyang[2], Shu Su[2], Sangsang Zhu[1], Qiaoli Chen[1], Yang Sheng[1], Liang Chen[1], Hong Wang[3], David G. Campbell [4], Carol MacKintosh[5], Zhongzhou Yang[1], Kunfu Ouyang[3], Hong Yu Wang[2,8✉] & Shuai Chen [1,6,8✉]

Diabetic cardiomyopathy is a progressive disease in diabetic patients, and myocardial insulin resistance contributes to its pathogenesis through incompletely-defined mechanisms. Striated muscle preferentially expressed protein kinase (SPEG) has two kinase-domains and is a critical cardiac regulator. Here we show that SPEG is phosphorylated on $Ser^{2461}$/$Ser^{2462}$/$Thr^{2463}$ by protein kinase B (PKB) in response to insulin. PKB-mediated phosphorylation of SPEG activates its second kinase-domain, which in turn phosphorylates sarcoplasmic/endoplasmic reticulum calcium-ATPase 2a (SERCA2a) and accelerates calcium re-uptake into the SR. Cardiac-specific deletion of PKBα/β or a high fat diet inhibits insulin-induced phosphorylation of SPEG and SERCA2a, prolongs SR re-uptake of calcium, and impairs cardiac function. Mice bearing a $Speg^{3A}$ mutation to prevent its phosphorylation by PKB display cardiac dysfunction. Importantly, the $Speg^{3A}$ mutation impairs SERCA2a phosphorylation and calcium re-uptake into the SR. Collectively, these data demonstrate that insulin resistance impairs this PKB-SPEG-SERCA2a signal axis, which contributes to the development of diabetic cardiomyopathy.

[1] State Key Laboratory of Pharmaceutical Biotechnology, Department of Cardiology, Nanjing Drum Tower Hospital, The Affiliated Hospital of Nanjing University Medical School, Model Animal Research Center, Nanjing University, 210061 Nanjing, China. [2] MOE Key Laboratory of Model Animal for Disease Study, Model Animal Research Center, Nanjing University, 210061 Nanjing, China. [3] Key Laboratory of Chemical Genomics, School of Chemical Biology and Biotechnology, Peking University, 518055 Shenzhen, China. [4] MRC Protein Phosphorylation and Ubiquitylation Unit, School of Life Sciences, University of Dundee, Dundee DD1 5EH Scotland, UK. [5] Division of Cell and Developmental Biology, School of Life Sciences, University of Dundee, Dundee DD1 5EH Scotland, UK. [6] Nanjing Biomedical Research Institute, Nanjing University, 210061 Nanjing, China. [7] These authors contributed equally: Chao Quan, Qian Du. [8] These authors jointly supervised this work: Hong Yu Wang, Shuai Chen. ✉email: wanghy@nicemice.cn; schen6@163.com

Type 2 diabetes (T2D) has become prevalent world-wide in the past few decades, and much of the high mortality rate in patients is due to heart disease[1]. Diabetic cardiomyopathy is a progressive disease, independent of coronary artery disease and hypertension, which begins early after the onset of diabetes and can eventually lead to heart failure in diabetic patients[2]. Although diabetic cardiomyopathy has been well recognized in recent years, the pathophysiological mechanisms of this disease are incompletely understood.

Multiple factors including structural, functional and metabolic changes in the cardiomyocytes may contribute to the development of diabetic cardiomyopathy[3,4]. In particular, insulin resistance, which is characteristic of type 2 diabetes, is involved in the pathogenesis of diabetic cardiomyopathy even when it only occurs locally in the heart[5]. Myocardial insulin resistance not only perturbs metabolism in the cardiomyocytes[6], but also causes mitochondrial dysfunction and oxidative stress in these cells[7]. Moreover, myocardial insulin resistance can impair calcium homeostasis through undefined mechanisms, which also contributes to cardiomyocyte dysfunction in diabetic cardiomyopathy[4,8,9]. Insulin signaling is initiated through binding of insulin to its receptor, which consequently activates the phosphatidylinositol 3-kinase (PI 3-kinase)–protein kinase B (PKB, also known as Akt) pathway[10]. However, it is not clear whether this insulin–PKB pathway directly regulates calcium homeostasis in cardiomyocytes, or solely exerts its effect indirectly via metabolic changes. Accumulative evidence show that an excess risk for heart failure persists in type 2 diabetic patients despite an optimal glycemic control[11], heightening the need to decipher the molecular mechanism regulating calcium homeostasis by the insulin–PKB pathway.

The striated muscle preferentially expressed protein kinase (SPEG) is a member of the MLCK subgroup of CaMK Ser/Thr protein kinase family, and plays a critical role in regulating cardiac development and function[12]. It regulates the cardiomyocyte cytoskeleton in the developing heart, and its deficiency causes dilated cardiomyopathy during embryo development and results in neonatal death in mice[12]. Importantly, homozygous and compound-heterozygous SPEG mutations are also associated with dilated cardiomyopathy in human patients[13]. SPEG has two serine/threonine (Ser/Thr) kinase (SK) domains in its C-terminal part, referred to as SK1 and SK2[14]. A few possible substrates for SPEG have been identified, including MTM3[12] and junctophilin-2 (JPH2)[15] that regulate cytoskeleton and t-tubule function, respectively. In a recent study, we identified sarcoplasmic/endoplasmic reticulum calcium ATPase 2a (SERCA2a) as a substrate for SPEG[16]. SERCA2a is an important ATPase for reuptake of calcium into the sarcoplasmic reticulum (SR) in cardiomyocytes during muscle relaxation[17]. We further found that the SK1 of SPEG was responsible for JPH2 phosphorylation while the SK2 could phosphorylate $Thr^{484}$ on SERCA2a. Phosphorylation of SERCA2a by SPEG promotes oligomerization of the $Ca^{2+}$ pump and increases reuptake of calcium into the SR[16]. Despite the importance of SPEG, it remains unknown how this protein, particularly the activities of its two kinase domains are regulated in the heart.

In this study, we identify SPEG as a protein that is phosphorylated by PKB in response to insulin in the heart. Phosphorylation of SPEG by PKB activates its SK2, which in turn phosphorylates SERCA2a. We utilize genetically-modified mouse models and their derived cardiomyocytes to demonstrate that impairment of this PKB−SPEG signaling nexus may contribute to the development of diabetic cardiomyopathy.

## Results

### PKBα/β deletion impairs heart function and SR $Ca^{2+}$ reuptake.
To investigate whether PKB regulates calcium homeostasis in cardiomyocytes, we deleted PKBα and β in mouse heart by mating a PKBα$^{f/f}$;PKBβ$^{−/−}$ mouse with a Myh6-MerCreMer (MCM) mouse[18]. Before tamoxifen induction, only PKBβ was deleted in the heart of PKBα$^{f/f}$;MCM;PKBβ$^{−/−}$ mice whereas expression of PKBα and γ was normal (Supplementary Fig. 1a). After treatment with tamoxifen, the level of PKBα protein was also markedly decreased, while PKBγ was substantially increased, in the heart of PKBα$^{f/f}$;MCM;PKBβ$^{−/−}$ mice as compared to the heart of PKBα$^{f/f}$;PKBβ$^{−/−}$ controls (Fig. 1a). Furthermore, we found that PKBα protein was decreased by ~80% in primary cardiomyocytes from tamoxifen-treated PKBα$^{f/f}$;MCM;PKBβ$^{−/−}$ mice as compared to that in control cardiomyocytes from PKBα$^{f/f}$;PKBβ$^{−/−}$ mice (Supplementary Fig. 1b–c). Insulin-induced phosphorylation of PKB substrates AS160 and GSK3 was blunted in the heart of PKBα$^{f/f}$;MCM;PKBβ$^{−/−}$ mice (Supplementary Fig. 1d). Both cardiomyocyte sizes and heart to body weight ratio were unaltered in the tamoxifen-treated PKBα$^{f/f}$;MCM;PKBβ$^{−/−}$ mice as compared to PKBα$^{f/f}$;PKBβ$^{−/−}$ controls (Supplementary Fig. 1e–f). Cardiac function was comparable in the PKBα$^{f/f}$;MCM;PKBβ$^{−/−}$ and PKBα$^{f/f}$;PKBβ$^{−/−}$ mice before tamoxifen induction, but was significantly decreased in the PKBα$^{f/f}$;MCM;PKBβ$^{−/−}$ mice from 2 weeks after tamoxifen treatment (Fig. 1b–c). Concurrently, the diameters and volumes of left ventricle (LV) became enlarged with thinner anterior and posterior walls in the PKBα$^{f/f}$;MCM;PKBβ$^{−/−}$ mice under both systolic and diastolic conditions (Fig. 1d–e, Supplementary Fig. 1g–l), suggesting that deletion of PKBα/β in adult hearts causes dilated cardiomyopathy. We confirmed that tamoxifen induction did not alter cardiac function in the MCM;PKBβ$^{−/−}$ mice as compared to PKBβ$^{-/-}$ controls (Supplementary Fig. 2). Cardiac dilation of the tamoxifen-treated PKBα$^{f/f}$;MCM;PKBβ$^{−/−}$ heart was associated with up-regulation of cardiac remodeling, cell apoptosis and fibrosis but normal expression of myofilament components such as tropomyosin-3 and troponin I (Supplementary Fig. 3a–e, Supplementary Fig. 4a–c). As a consequence of cardiac dysfunction, the PKBα$^{f/f}$;MCM;PKBβ$^{−/−}$ mice started to die from 3 weeks on after tamoxifen induction (Supplementary Fig. 3f).

We isolated primary cardiomyocytes from these two genotypes of mice at 4 weeks after tamoxifen induction, and measured $Ca^{2+}$ transients. The peaks of $Ca^{2+}$ transients were unaltered in the cardiomyocytes from PKBα$^{f/f}$;MCM;PKBβ$^{−/−}$ mice (Fig. 1f). Interestingly, the full duration at half maximum (FDHM) and time constant Tau of $Ca^{2+}$ transients were both significantly increased in the cardiomyocytes from PKBα$^{f/f}$;MCM;PKBβ$^{−/−}$ mice as compared to those in the PKBα$^{f/f}$;PKBβ$^{-/-}$ control cells (Fig. 1f). Together, these data show that PKB regulates calcium reuptake into SR of cardiomyocytes.

### Identification of SPEG as a PKB target.
To gain insights into how PKB regulates calcium reuptake into SR of cardiomyocytes, we used a generic antibody recognizing phospho-Akt substrates (named the PAS antibody) to identify possible PKB substrates in the heart as previously reported[19]. PAS-reactive signals were increased on a number of proteins in the heart of mice upon insulin stimulation (Fig. 2a). These phosphoproteins were immunoprecipitated from heart lysates using the PAS antibody and identified via mass-spectrometry (Fig. 2b, Supplementary Data 1). Among the 78 proteins identified, a few are known PKB substrates, including AS160, RalGAPα1, RalGAPα2, and TSC2, and their abundance in the PAS immunoprecipitates was substantially higher upon insulin stimulation than in the basal state in wild-type mouse heart (Fig. 2c). Among potential novel insulin targets was the protein kinase SPEG that is a key regulator of SERCA2a (Supplementary Data 1). We confirmed that the

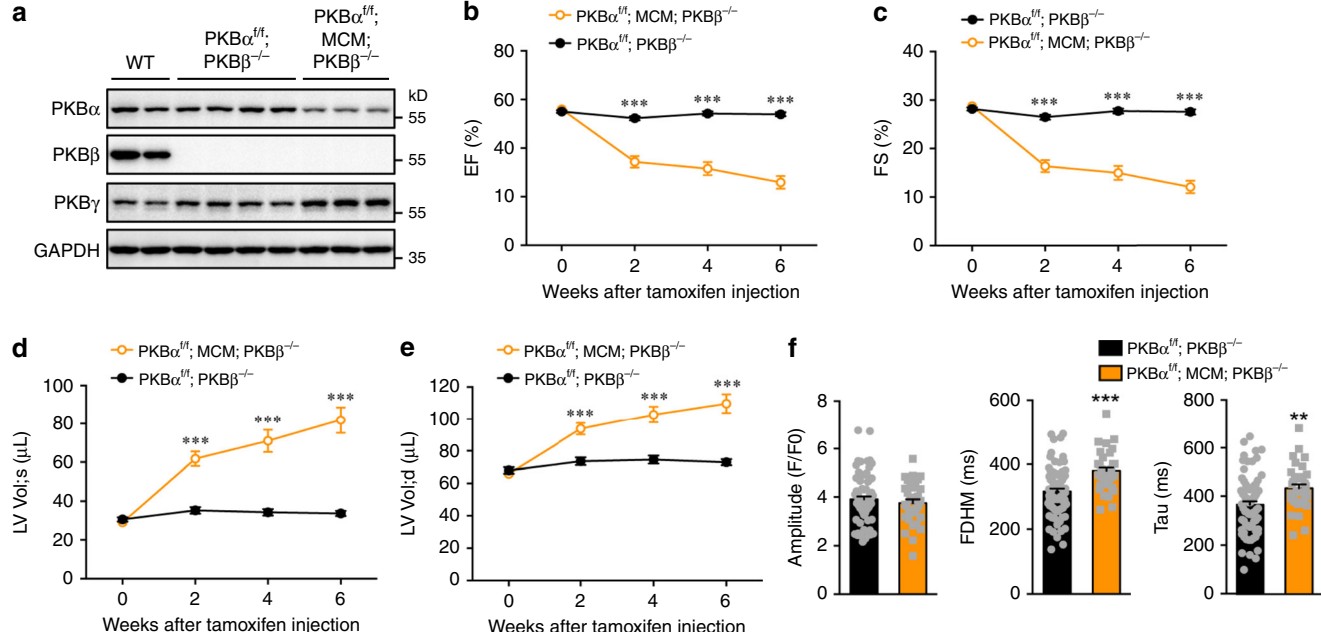

**Fig. 1 Effects of PKBα/β deletion in the heart on cardiac function and calcium transients in primary cardiomyocytes. a** PKBα, β and γ protein expression in the heart of male PKBα$^{f/f}$;MCM;PKBβ$^{-/-}$ mice at 8 weeks after tamoxifen induction. **b–e** Ejection fraction (EF) (**b**), fractional shortening (FS) (**c**), systolic left ventricular volume (LV Vol;s) (**d**), and diastolic left ventricular volume (LV Vol;d) (**e**) were measured via echocardiography in the male PKBα$^{f/f}$; PKBβ$^{-/-}$ and PKBα$^{f/f}$;MCM;PKBβ$^{-/-}$ mice before and after tamoxifen induction. $n = 14$ (0 week), 12 (2 week), 12 (4 week), and 12 (6 week) for PKBα$^{f/f}$; PKBβ$^{-/-}$ mice. $n = 12$ (0 week), 9 (2 week), 8 (4 week), and 7 (6 week) for PKBα$^{f/f}$;MCM;PKBβ$^{-/-}$ mice. $p = 0.147$ (0 week), 1.60e-7 (2 week), 2.15e-8 (4 week) and 5.33e-10 (6 week) for EF. $p = 0.165$ (0 week), 8.56e-8 (2 week), 1.16e-8 (4 week), and 3.03e-10 (6 week) for FS. $p = 0.229$ (0 week), 1.05e-6 (2 week), 9.84e-7 (4 week), and 6.36e-8 (6 week) for LV Vol;s. $p = 0.387$ (0 week), 9.36e-5 (2 week), 2.70e-5 (4 week), and 1.23e-6 (6 week) for LV Vol;d. **f** Calcium transients elicited by electrical stimulation in primary cardiomyocytes isolated from the male PKBα$^{f/f}$;PKBβ$^{-/-}$ and PKBα$^{f/f}$;MCM;PKBβ$^{-/-}$ mice at 4 weeks after tamoxifen induction. Quantitation of amplitude, full duration at half maximum (FDHM) and time constant Tau of calcium transients was shown. 76 cells from 3 PKBα$^{f/f}$;PKBβ$^{-/-}$ mice and 34 cells from 2 PKBα$^{f/f}$;MCM;PKBβ$^{-/-}$ mice were analyzed. $p = 0.446$ (amplitude), 5.96e-5 (FDHM), and 3.99e-3 (tau). The data are given as the mean ± SEM. Statistical analyses for **b–f** were carried out using two-sided $t$-test. Two-asterisk indicates $p < 0.01$, and three-asterisk indicates $p < 0.001$. Source data are provided as a Source Data file.

abundance of SPEG in the PAS immunoprecipitates was increased in response to insulin stimulation in wild-type mouse heart similarly to the above-mentioned known PKB substrates (Fig. 2d). Furthermore, when a GFP-SPEG fusion protein was expressed in cells, insulin stimulated its PAS-reactive phosphorylation, which could be inhibited by pre-treatment with a PI 3-kinase inhibitor, PI-103, or with a PKB inhibitor, Akti1/2 (Fig. 2e–f). Deficiency of PKBα/β caused insulin resistance in mouse heart, and the abundance of their substrates including AS160, RalGAPα1, RalGAPα2, and TSC2 in the PAS immunoprecipitates was substantially lower for PKBα/β-deficient mouse heart relative to wild-type mouse heart upon insulin stimulation (Fig. 2c). Insulin resistance due to PKBα/β deficiency also diminished the presence of SPEG in the PAS immunoprecipitates (Fig. 2d). Taken together, these data show that SPEG is a target of the insulin−PI 3-kinase−PKB signaling pathway in the heart, and insulin resistance impaired its PAS-reactive phosphorylation in the heart.

### PKB phosphorylates SPEG on Ser$^{2461}$/Ser$^{2462}$/Thr$^{2463}$ residues.
To elucidate biochemical aspects of regulation of SPEG by insulin, GFP-SPEG was immunoprecipitated from cell lysates, and phosphorylated residues that cluster in two regions on SPEG were identified via mass-spectrometry (Supplementary Fig 5, Supplementary Data 2). By fragmentation analysis, we identified PAS-reactive signals within a region spanning Pro$^{2227}$ to Ser$^{2583}$ (Fig. 3a–c). Within this region, a small cluster of serine/threonine residues namely Ser$^{2461}$/Ser$^{2462}$/Thr$^{2463}$ conform to the PKB

consensus motif (RXRXXpS/T), and their mutation to non-phosphorylatable alanine abolished insulin-stimulated PAS-reactive signals (Fig. 3d–f). Furthermore, the purified SPEG$^{P2227-S2583}$ fragment could be phosphorylated by PKB in vitro, and the triple-alanine substitution of Ser$^{2461}$/Ser$^{2462}$/Thr$^{2463}$ prevented its PAS-reactive phosphorylation by PKB (Fig. 3g). Together, these data showed that SPEG is a PKB substrate that is phosphorylated upon insulin stimulation, and PKB phosphorylates SPEG on the cluster of serine/threonine residues including Ser$^{2461}$/Ser$^{2462}$/Thr$^{2463}$ that could be detected by the PAS antibody.

### PAS-reactive phosphorylation of SPEG activates its SK2.
The cluster of serine/threonine residues Ser$^{2461}$/Ser$^{2462}$/Thr$^{2463}$ are located between the SK1 and SK2, and we next investigated whether PAS-reactive phosphorylation regulates these two kinase domains of SPEG. It has been reported that SPEG can regulate t-tubule function probably through phosphorylating JPH2[15]. In agreement with this report, we also found that tamoxifen-induced deletion of SPEG impaired t-tubule function in the heart as indicated by decreased TT-power (Supplementary Fig. 6a–b). In a recent study, we showed that SK1 but not SK2 of SPEG is required for JPH2 phosphorylation[16]. Consistent with its phosphorylation by SK1, JPH2 was not phosphorylated by SPE-G$^{Asp1746Gly}$ mutant protein, in which Asp$^{1746}$ was mutated to glycine to inactivate SK1. In contrast, mutant SPEG in which Ser$^{2461}$, Ser$^{2462}$, and Thr$^{2463}$ were replaced with non-phosphorylatable alanine (SPEG$^{3A}$ mutant protein) phosphorylated JPH2 at a rate comparable to that of wild-type SPEG

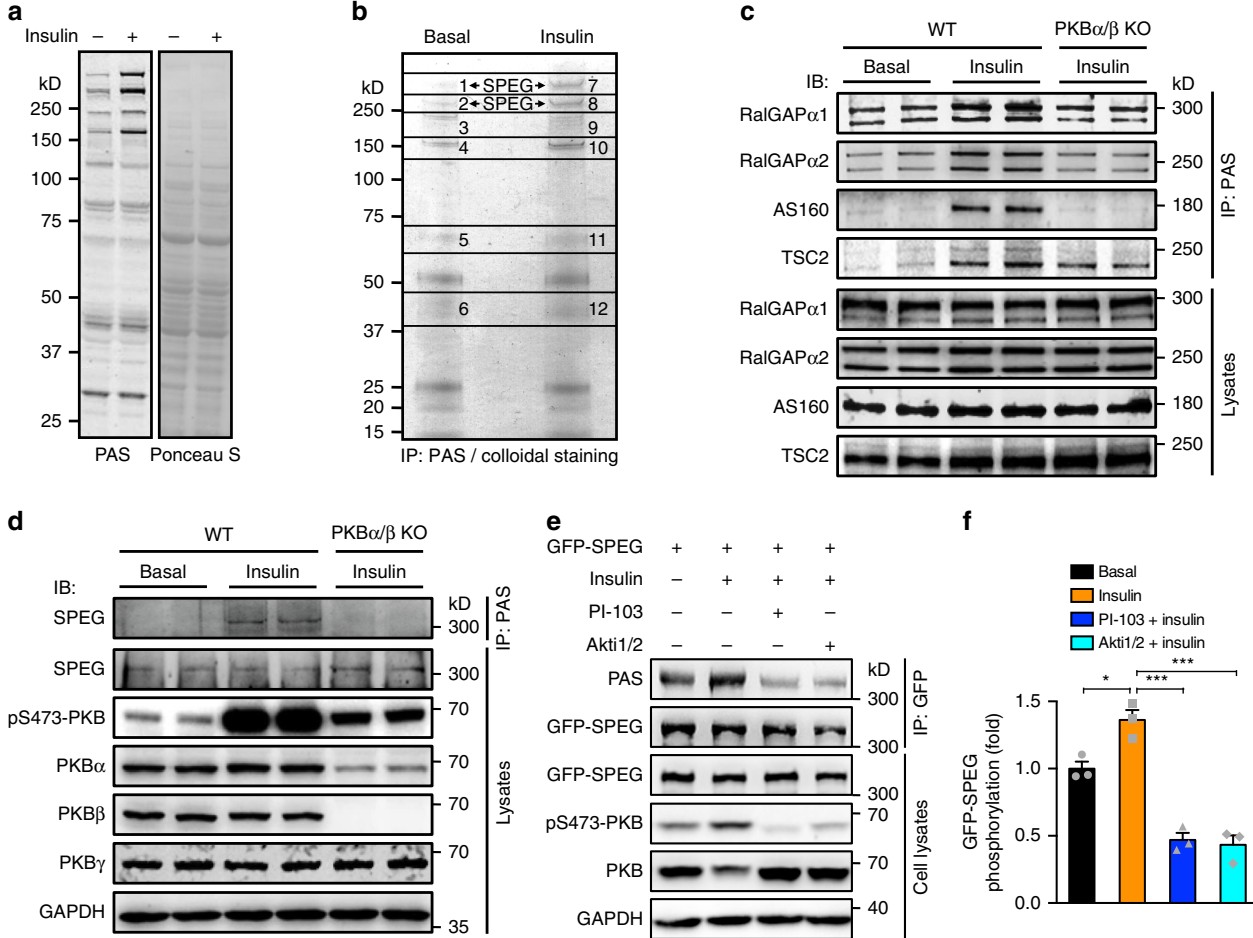

**Fig. 2 Identification of SPEG as a phosphoprotein regulated by the insulin–PKB pathway. a** PAS-reactive protein phosphorylation in lysates of hearts from mice that were intraperitoneally injected with or without insulin. **b** Immunoprecipitation of potential PKB substrates from lysates of mouse hearts stimulated with or without insulin using the PAS antibody. Immunoprecipitated proteins were identified via mass-spectrometry and summarized in Supplementary Data 1. **c** Immunoprecipitation of known PKB substrates from lysates of WT and PKBα/β KO (PKBα[f/f];αMHC-Cre;PKBβ[−/−]) mouse hearts stimulated with or without insulin using the PAS antibody. RalGAPα1, RalGAPα2, AS160, and TSC2 in the PAS immunoprecipitates were detected with the corresponding antibodies via western blot. **d** Immunoprecipitation of phosphorylated SPEG from lysates of WT and PKBα/β KO (PKBα[f/f];αMHC-Cre; PKBβ[−/−]) mouse hearts stimulated with or without insulin using the PAS antibody. SPEG in the PAS immunoprecipitates and heart lysates was detected via western blot. **e, f.** Full-length SPEG was tagged with GFP and expressed in HEK293 cells that were treated with the PI 3-kinase inhibitor PI-103 and PKB inhibitor Akti1/2 in the presence or absence of insulin. PAS-reactive phosphorylation of SPEG was detected on immunoprecipitated SPEG using the PAS antibody. **e** representative blots. **f** Quantitation of PAS-reactive phosphorylation of GFP-SPEG. $n = 3$. The data are given as the mean ± SEM. Statistical analysis was carried out using one-way ANOVA. One-asterisk indicates $p < 0.05$, and three-asterisk indicates $p < 0.001$. Source data are provided as a Source Data file.

(Supplementary Fig. 6c). These data suggest that PAS-reactive phosphorylation of SPEG does not affect its SK1 activity.

The SK2, but not SK1, of SPEG can phosphorylate SERCA2a[16]. In agreement with this report, SPEG[Asp3098Ala] mutant protein (in which Asp[3098] was mutated to alanine to inactivate SK2) was unable to phosphorylate SERCA2a (Fig. 4a, b). Importantly, the SPEG[3A] mutant protein also failed to phosphorylate SERCA2a (Fig. 4a, b), suggesting that PAS-reactive phosphorylation of SPEG is required for SK2 activation. Phosphorylation of SERCA2a by SPEG increased oligomerization of SERCA2a[16]. Therefore, we further investigated whether SPEG[3A] mutant protein could increase oligomerization of SERCA2a. Three lines of evidence showed that SPEG[3A] mutant protein was unable to increase oligomerization of SERCA2a (Fig. 4c–h). First, unlike wild-type SPEG, but similar to SPEG[Asp3098Ala] mutant protein, co-expression of SERCA2a with SPEG[3A] mutant protein failed to increase levels of high molecular weight SERCA2a (~ 300 kDa)

(Fig. 4c, d). Second, a FRET-based assay showed that wild-type SPEG, but not SPEG[3A] and SPEG[Asp3098Ala] mutants could increase the efficiency of FRET between CFP-SERCA2a and YFP-SERCA2a (Fig. 4e, f). Third, co-expression of wild-type SPEG, but not SPEG[3A] and SPEG[Asp3098Ala] mutant proteins, enhanced co-immunoprecipitation of HA-SERCA2a with Flag-SERCA2a (Fig. 4g–h). These results were in agreement with SPEG[3A] mutant protein being unable to phosphorylate SERCA2a. Moreover, we found that the FDHM and Tau of Ca[2+] transients were significantly larger when SERCA2a was co-expressed in HEK293 cells with SPEG[3A] or SPEG[Asp3098Ala] mutant proteins compared with SPEG wild-type protein (Fig. 4i, j), consistent with a critical role for SPEG-stimulated phosphorylation and oligomerization of SERCA2a in activation of this Ca[2+] pump. In agreement, overexpression of SPEG in primary neonatal rat cardiomyocytes decreased the FDHM and Tau of Ca[2+] transients and increased their peaks, suggesting acceleration of SR calcium reuptake in

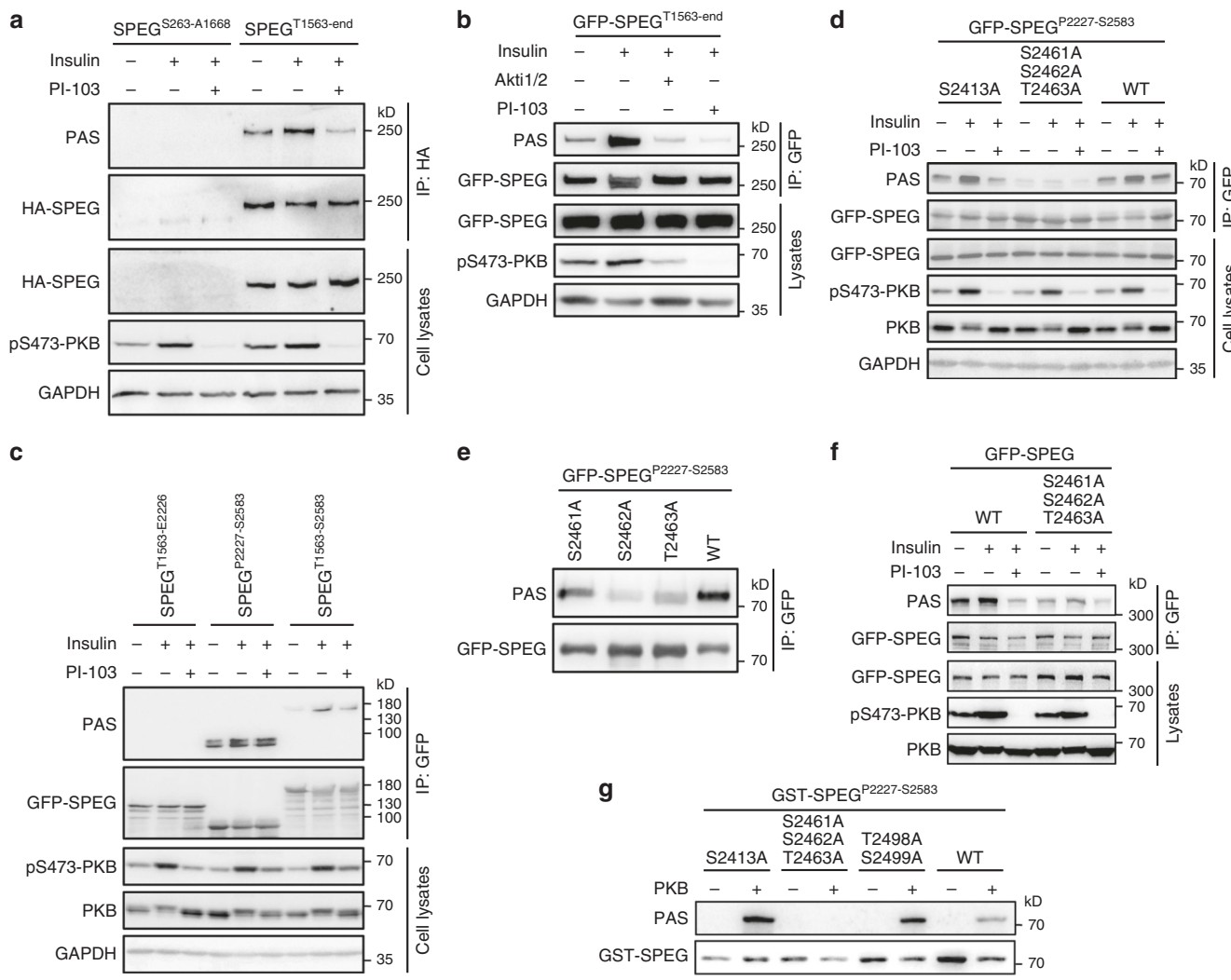

**Fig. 3 Identification of PKB-mediated phosphorylation sites on SPEG. a–c** SPEG fragments were expressed in HEK293 cells that were treated with indicated inhibitors and insulin. PAS-reactive phosphorylation of SPEG was detected on immunoprecipitated SPEG using the PAS antibody. **d, e** GFP-SPEG[P2227-S2583] WT and mutant proteins were expressed in HEK293 cells. PAS-reactive phosphorylation of SPEG was detected on immunoprecipitated proteins using the PAS antibody. **f** GFP-SPEG WT and mutant proteins were expressed in HEK293 cells. PAS-reactive phosphorylation of SPEG was detected on immunoprecipitated proteins using the PAS antibody. **g** In vitro phosphorylation of GST-SPEG[P2227-S2583] WT and mutant recombinant proteins by PKB. Source data are provided as a Source Data file.

these cells (Fig. 4k). Such effects were not observed when SPEG[3A] mutant protein was expressed in neonatal rat cardiomyocytes (Fig. 4k).

Taken together, these data suggest that PAS-phosphorylation of SPEG activates SK2 but not SK1.

**Insulin resistance impairs the SPEG–SERCA2a signaling nexus.** We next investigated how insulin resistance affects the SPEG–SERCA2a signaling nexus. We first employed PKBα/β-deficient cardiomyocytes as an insulin-resistant model. Insulin expectedly stimulated phosphorylation of PKB and SPEG in WT cardiomyocytes (Fig. 5a, b). Importantly, Thr[484] phosphorylation of SERCA2a was significantly increased in WT cardiomyocytes in response to insulin (Fig. 5a, b). In contrast, insulin-stimulated phosphorylation of both SPEG and SERCA2a was blunted in PKBα/β-deficient cardiomyocytes (Fig. 5a, b). Furthermore, SERCA2a-Thr[484] phosphorylation was decreased in the heart of ad libitum PKBα[f/f];MCM;PKBβ[−/−] mice in which PAS-phosphorylation of SPEG was also diminished (Supplementary Fig. 4d, g). PLB phosphorylation was increased in these hearts

probably as a compensatory response (Supplementary Fig. 4a, b). It has been reported that PKB mediates β-AR (β-adrenergic receptor)-induced SR Ca[2+] leak via CaMKII-dependent phosphorylation of RyR2 under hypertrophic conditions[20]. We found that phosphorylation of RyR2 remained normal in the PKBα/β-deficient hearts (Supplementary Fig. 4a, b), suggesting that PKB might not be responsible for RyR2 phosphorylation under basal conditions. We then used insulin-resistant rat H9C2 cardiomyocytes induced with chronic insulin treatment as a second model. Insulin resistance in this model also significantly impaired insulin-stimulated phosphorylation of PKB, SPEG and SERCA2a (Fig. 5c, d). Lastly, we utilized mice fed with a high fat diet (HFD) as a third insulin-resistant model, in which decay of Ca[2+] transients was prolonged and cardiac function was impaired (Fig. 6a, c). As expected, insulin-stimulated PKB phosphorylation was inhibited in cardiomyocytes from HFD-fed mice as compared to that in cardiomyocytes from mice fed with a chow diet (CD) (Fig. 6d, e). Importantly, insulin-stimulated phosphorylation of SPEG and SERCA2a were lower in cardiomyocytes from HFD-fed mice than in cardiomyocytes from CD-fed mice (Fig. 6d, e).

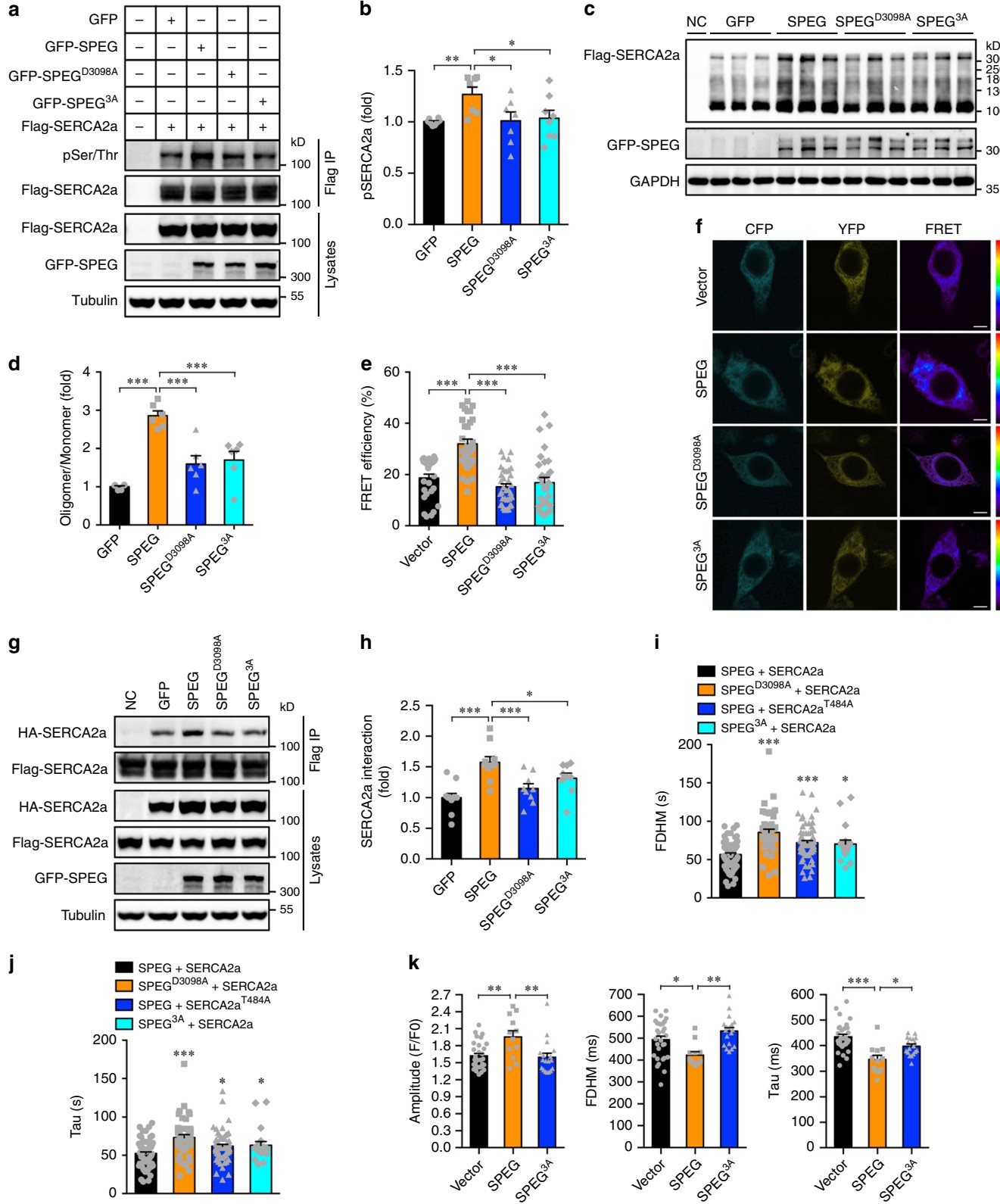

Together, these data demonstrate that insulin resistance impairs the insulin−PKB−SPEG−SERCA2a signaling axis in cardiomyocytes.

**Generation of transgenic mice bearing a *Speg*³ᴬ mutation**. To further delineate the in vivo function of PAS-reactive phosphorylation of SPEG, we generated a *Speg*³ᴬ-knockin mutant mouse in which Ser²⁴⁶¹, Ser²⁴⁶², and Thr²⁴⁶³ of SPEG were substituted with triple-alanine (Fig. 7a, Supplementary Fig. 7a, b). The *Speg*³ᴬ-knockin mice had normal blood glucose and lipids, and displayed similar tolerance to an oral glucose load as the wild-type littermates (Fig. 7b, c). The SPEG³ᴬ mutant protein was expressed at levels comparable to SPEG in hearts of wild-type mice (Fig. 7d). Phosphorylation of TSC2, mTOR and AS160 were

**Fig. 4 Effects of PKB-mediated phosphorylation of SPEG on the activity of its second kinase-domain. a, b** Flag-SERCA2a was co-expressed with GFP-SPEG WT or mutant proteins in HEK293 cells. After immunoprecipitation, phosphorylation of Flag-SERCA2a was detected using the pSer/Thr antibody. **a** representative blots. **b** Quantitation of SERCA2a phosphorylation. $n = 8$ (GFP and SPEG[3A]) and 7 (SPEG and SPEG[D3098A]). **c, d** Flag-SERCA2a was co-expressed with GFP-SPEG WT or mutant proteins in HEK293 cells. Oligomerization of SERCA2a was determined via western blot and subsequently quantified. Representative blots were shown in **c** and quantitative data shown in **d**. $n = 6$. E-F. CFP-SERCA2a and YFP-SERCA2a were co-expressed with HA-SPEG WT or mutant proteins in HEK293 cells. Inter-molecular interaction of SERCA2a was measured via FRET. E, quantitative data on FRET efficiency. **f** representative images for FRET. $n = 28$ (vector), 30 (SPEG), 35 (SPEG[D3098A]), and 29 (SPEG[3A]). Bars indicate 5 μm in length. **g, h** HA-SERCA2a was co-expressed with Flag-SERCA2a in the presence of GFP-SPEG WT and mutant proteins in HEK293 cells. Flag-SERCA2a was immunoprecipitated, and the abundance of HA-SERCA2a in the immunoprecipitates was detected via immunoblotting. **g** representative blots. **h** Quantitative data. $n = 11$ (GFP), 10 (SPEG), and 9 (SPEG[D3098A] and SPEG[3A]). **i, j** Calcium transients in HEK293 cells expressing mCherry-SERCA2a together with HA-SPEG WT or mutant proteins. Calcium transients were recorded using a confocal microscopy in cells that were stimulated with ATP. Full duration at half maximum (FDHM, **i**), and time constant Tau (**j**) of calcium transients were subsequently determined. $n = 68$ (SERCA2a WT + SPEG WT), 43 (SERCA2a WT + SPEG[D3098A]), 64 (SPEG + SERCA2a[T484A]), and 20 (SERCA2a WT + SPEG[3A]). **k** Calcium transients in neonatal rat cardiomyocytes expressing vector, mCherry-SPEG WT or mCherry-SPEG[3A] mutant proteins upon field stimulation. Amplitudes, FDHM and Tau of calcium transients were quantified from 30 (vector), 12 (mCherry-SPEG WT) or 18 (mCherry-SPEG[3A]) cells. The data are given as the mean ± SEM. Statistical analyses were carried out using one-way ANOVA. One-asterisk indicates $p < 0.05$, two-asterisk indicates $p < 0.01$, and three-asterisk indicates $p < 0.001$. Source data are provided as a Source Data file.

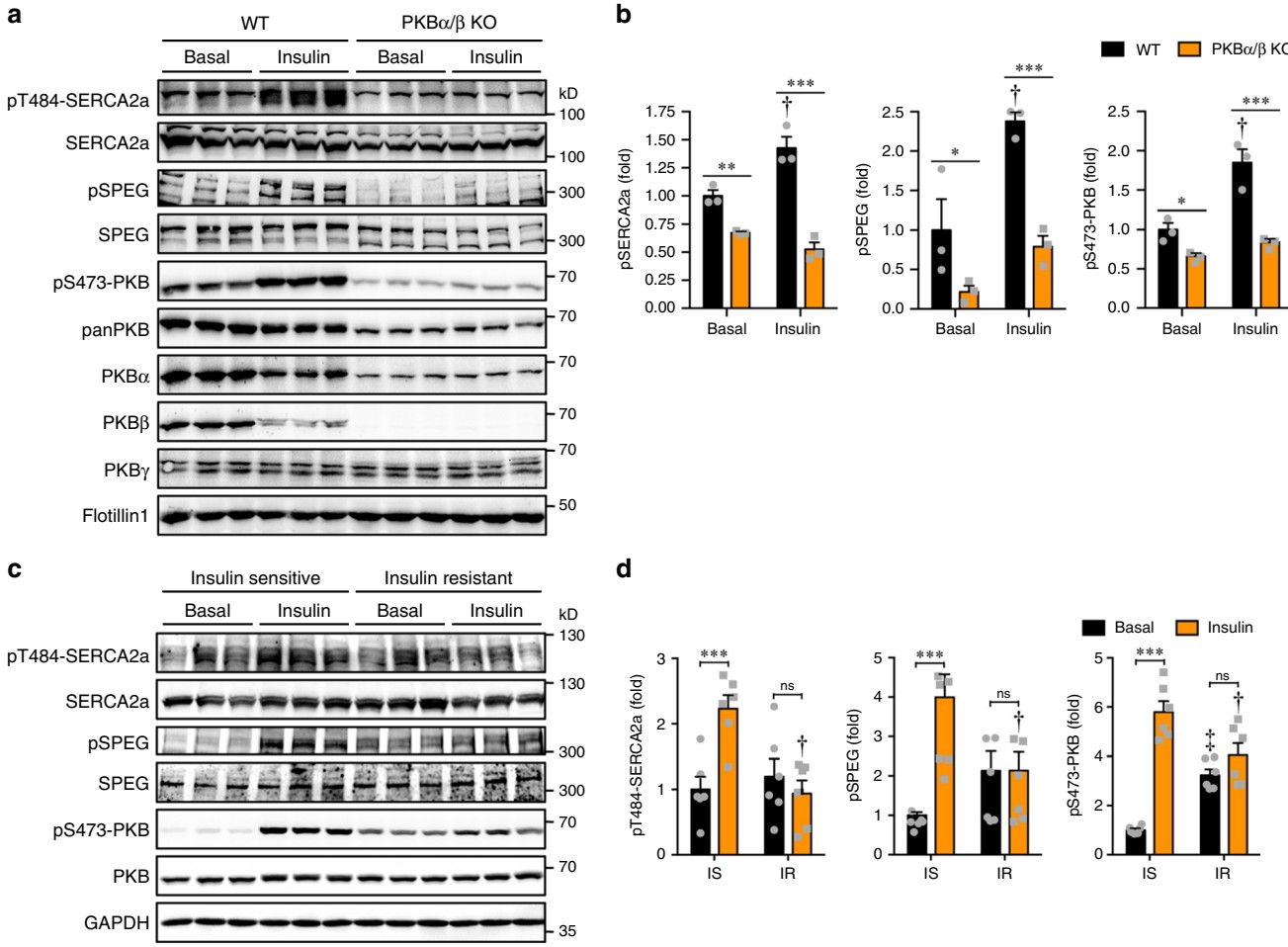

**Fig. 5 Phosphorylation of PKB-SPEG–SERCA2a axis in insulin-resistant cardiomyocytes. a, b** Phosphorylation of PKB, SPEG, and SERCA2a in WT or PKBα/β-KO primary cardiomyocytes stimulated with or without insulin. Phosphorylation of PKB, SPEG and SERCA2a was normalized with their respective total proteins, and quantitative data were shown in **b**. $n = 3$. **c, d**. Phosphorylation of PKB, SPEG and SERCA2a in insulin-sensitive or resistant H9C2 cardiomyocytes stimulated with or without insulin. H9C2 cardiomyocytes were differentiated for 7 days before insulin resistance was induced by prolonged treatment with 300 nM insulin for 24 h. Cells were then stimulated with or without insulin for 30 min. Representative blots were shown in **c**. Phosphorylation of PKB, SPEG and SERCA2a was normalized with their respective total proteins, and quantitative data were shown in **d**. $n = 6$. IS insulin sensitive, IR insulin-resistant. One-dagger (IS insulin vs IR insulin) indicates $p < 0.01$. One-diesis (IS basal vs IR basal) indicates $p < 0.001$. The data are given as the mean ± SEM. Statistical analyses were carried out using two-way ANOVA. One-asterisk indicates $p < 0.05$, two-asterisk indicates $p < 0.01$, and three-asterisk indicates $p < 0.001$. Source data are provided as a Source Data file.

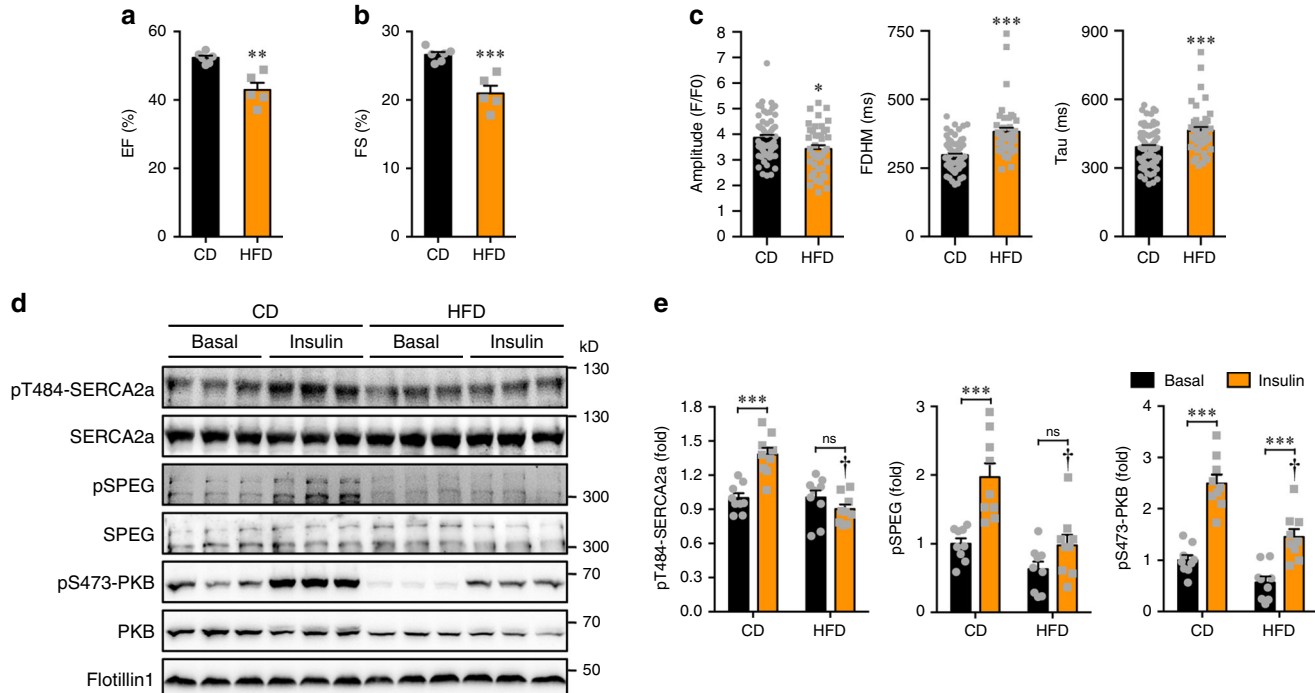

**Fig. 6 Effects of HFD on calcium homeostasis and phosphorylation of PKB-SPEG-SERCA2a axis in primary cardiomyocytes. a**, **b**. Ejection fraction (EF, **a**) and fractional shortening (FS, **b**) in male mice fed with chow diet (CD) or high fat diet (HFD) for 3 months. $n = 6$ (CD) and 5 (HFD). $p = 1.29e-3$ (EF) and $7.05e-4$ (FS). **c** Calcium transients in primary cardiomyocytes isolated from male mice fed with CD or HFD upon electrical stimulation. 64 cells from CD-fed mice and 42 cells from HFD-fed mice were analyzed. $p = 1.24e-2$ (amplitude), $1.79e-7$ (FDHM), and $2.52e-4$ (Tau). D-E. Phosphorylation of PKB, SPEG and SERCA2a in primary cardiomyocytes isolated from CD- or HFD-fed mice in response to insulin. Phosphorylation of PKB, SPEG and SERCA2a was normalized with their respective total proteins. Represent blots were shown in **d**, and quantitative data were shown in **e**. $n = 9$. The data are given as the mean ± SEM. Statistical analyses were carried out using two-sided $t$-test for **a–c**, and two-way ANOVA for **e**. One-asterisk indicates $p < 0.05$, two-asterisk indicates $p < 0.01$, and three-asterisk indicates $p < 0.001$. One-dagger (CD insulin vs HFD insulin) indicates $p < 0.001$. Source data are provided as a Source Data file.

unaltered in the heart of ad libitum $Speg^{3A}$-knockin mice (Supplementary Fig. 8a, b). Moreover, the $Speg^{3A}$-knockin mice also displayed normal phosphorylation of both PKB and of its substrate AS160 in response to insulin in the heart (Fig. 7d). Insulin-stimulated glucose uptake was comparable in $Speg^{3A}$-knockin and WT cardiomyocytes, and expression of key regulators for glucose and lipid metabolism were unchanged in the heart of $Speg^{3A}$-knockin mice (Supplementary Fig. 8c, g). As expected, insulin stimulation increased PAS-reactive phosphorylation of SPEG in the wild-type heart. Importantly, PAS-reactive phosphorylation of $SPEG^{3A}$ mutant protein was blunted in response to insulin treatment in the knockin heart (Fig. 7d). These data demonstrate the suitability of the $Speg^{3A}$-knockin mice and their derived cells for studying the specific in vivo and in vitro roles of PAS-reactive phosphorylation of SPEG.

**Impairment of SR Ca$^{2+}$ reuptake in $Speg^{3A}$ cardiomyocytes.** We next investigated whether the $Speg^{3A}$-knockin mutation affected calcium reuptake into the SR in cardiomyocytes. The knockin mice had normal sizes of cardiomyocytes (Supplementary Fig. 9a). Expression of SERCA2a, phospholamban and ryanodine receptor 2 (RyR2), and phosphorylation of phospholamban and RyR2 were unaltered in their hearts (Fig. 8a, Supplementary Fig. 9b, c). Unlike the WT cardiomyocytes where insulin stimulated Thr$^{484}$ phosphorylation of SERCA2a, the knockin cardiomyocytes could no longer respond to insulin to increase SERCA2a phosphorylation (Fig. 8a, b). Moreover, Thr$^{484}$ phosphorylation of SERCA2a was also significantly lower in the heart of ad libitum $Speg^{3A}$-knockin mice than in the control heart (Fig. 8c, d). SERCA2a oligomerization was also markedly

decreased in the heart of $Speg^{3A}$-knockin mice (Fig. 8e, f). These changes did not affect the ATPase activity of SERCA2a, but caused a significant inhibition of its Ca$^{2+}$ transport activity (Fig. 8g, h), which is reminiscent of effects on SERCA2a imposed by SPEG deficiency in the heart [16]. In agreement with the inhibition of Ca$^{2+}$ transport activity of SERCA2a, the FDHM and Tau of calcium transients elicited by electric stimulation were both significantly increased in the cardiomyocytes from $Speg^{3A}$-knockin mice (2-month and 7-month of age) as compared to those in control cardiomyocytes from wild-type littermates (Fig. 8i, j, Supplementary Fig. 10a). The peaks of Ca2+ transiets were normal in $Speg^{3A}$-knockin cardiomyocytes from 2-month-old mice, but became significantly lower in $Speg^{3A}$-knockin cardiomyocytes from 7-month-old mice than in WT control cells (Supplementary Fig. 10a). The fequency of spontaneous calcium sparks was normal in the $Speg^{3A}$-knockin cardiomyocytes (Fig. 8k). Again, the FDHM and Tau of calcium sparks were significantly larger in the $Speg^{3A}$-knockin cardiomyocytes than those in wild-type control cardiomyocytes (Fig. 8k). In agreement with SPEG$^{3A}$ mutant protein being able to phosphorylate JPH2 (Supplementary Fig. 6c), TT-power remained normal in the $Speg^{3A}$-knockin cardiomyocytes (Supplementary Fig. 10b, c), suggesting that t-tubule function was most likely unaffected in the $Speg^{3A}$-knockin heart. Together, these data demonstrate that the $Speg^{3A}$-knockin mutation impaired calcium reuptake via SERCA2a into the SR.

**The $Speg^{3A}$-knockin mutation impaired heart function in mice.** We next performed echocardiography to investigate in vivo effects of the $Speg^{3A}$-knockin mutation on cardiac function.

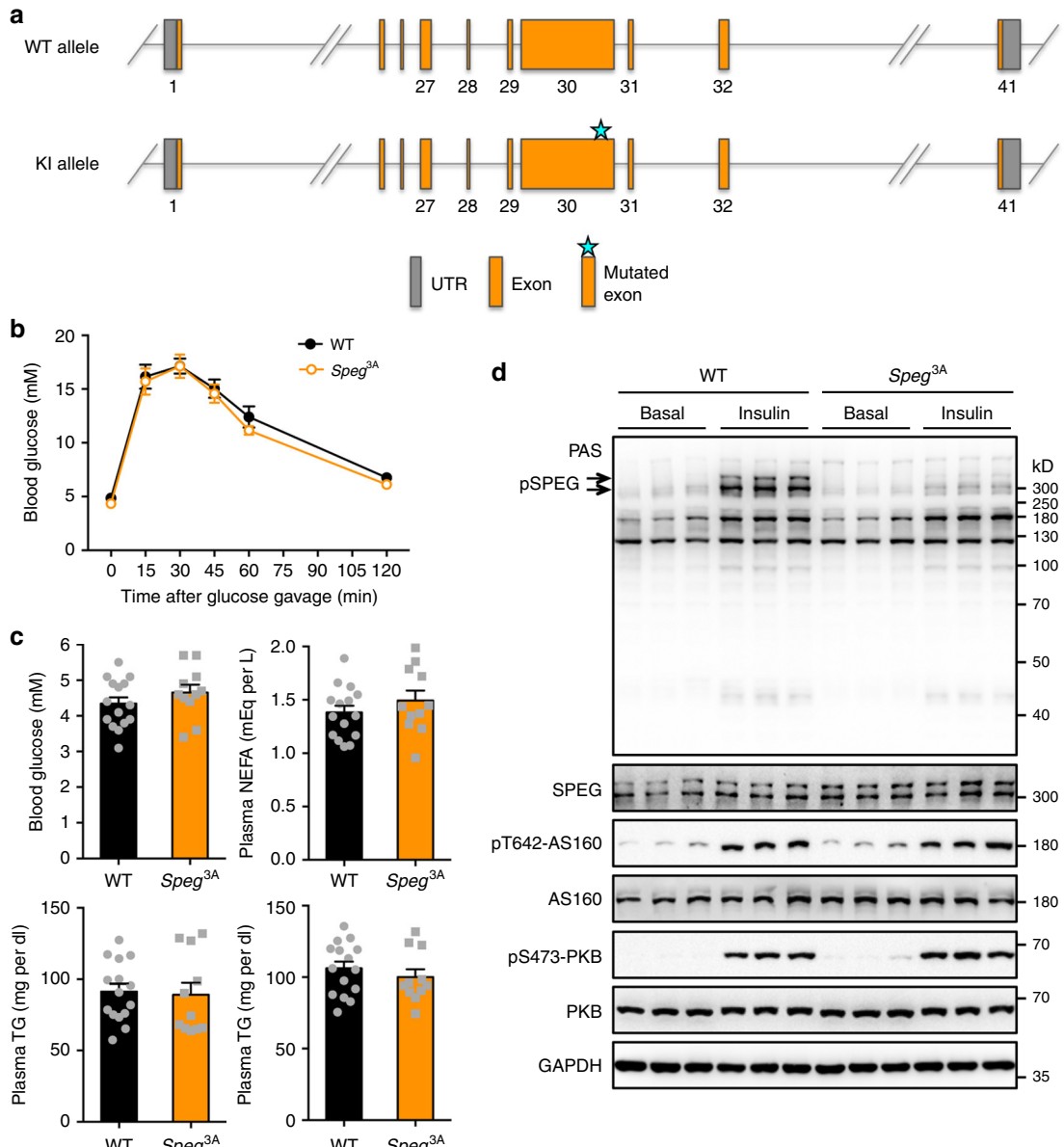

**Fig. 7 Generation and basic characterization of the *Speg*[3A]-knockin mice. a** Diagram of strategy for generation of the *Speg*[3A]-knockin mice. The cluster of serine/threonine residues Ser[2461]/Ser[2462]/Thr[2463] (the surrounding sequence is LAVRRRLsstLERL, Ser[2461]/Ser[2462]/Thr[2463] shown in lower case, and numbering is according to NP_031489.4) on SPEG was substituted to alanine by point mutagenesis. **b** Oral glucose tolerance test in the male *Speg*[3A]-knockin mice and wild-type littermates (9-month-old). $n = 5$ (WT) and 6 (*Speg*[3A]). $p = 0.050$ (0 min), 0.799 (15 min), 0.996 (30 min), 0.718 (45 min), 0.244 (60 min), and 0.134 (120 min). **c** Levels of glucose, non-esterified fatty acids (NEFA), triglycerides (TG) and total cholesterol (TC) were determined in the blood of male *Speg*[3A]-knockin mice and wild-type littermates (8-month-old) after an overnight fast. $n = 15$ (WT) and 11 (*Speg*[3A]). $p = 0.270$ (blood glucose), 0.317 (plasma NEFA), 0.819 (plasma TG), and 0.412 (plasma TC). **d** SPEG expression and PAS-reactive phosphorylation in the heart of WT and *Speg*[3A]-knockin mice in response to insulin. The data are given as the mean ± SEM. Statistical analyses were carried out using two-sided *t*-test. Source data are provided as a Source Data file.

Interestingly, both ejection fraction and fractional shortening were significantly lower in the hearts of young *Speg*[3A] knockin mice (~2-month-old), and further decreased when they became older (~5-month-old) (Fig. 9a, Supplementary Fig. 11). The *Speg*[3A]-knockin hearts did not display symptoms of cardiac hypertrophy; however, they progressed to become dilated with enlarged left ventricle (LV) volumes and thinner LV walls (Fig. 9a, Supplementary Fig. 11). The end-systolic LV volumes were increased in the young *Speg*[3A]-knockin mice (~2-month-old), and continued to expand when these animals became older (~5-month-old) (Fig. 9a, Supplementary Fig. 11). The end-

diastolic LV volumes were unaltered in the young *Speg*[3A]-knockin mice (~2-month-old); however, they also became significantly enlarged when the knockin male animals got older (~5-month-old) (Fig. 9a, Supplementary Fig. 11). The *Speg*[3A]-knockin mutation caused no cardiac remodeling, cell apoptosis or fibrosis, and did not alter expression of myofilament components including tropomyosin-3 and troponin I, within the experimental period (Supplementary Figs. 9 and 12). Together, these data demonstrate that PAS-reactive phosphorylation of SPEG plays a critical role in the heart at the downstream of insulin−PKB pathway, and links insulin signaling with cardiac function.

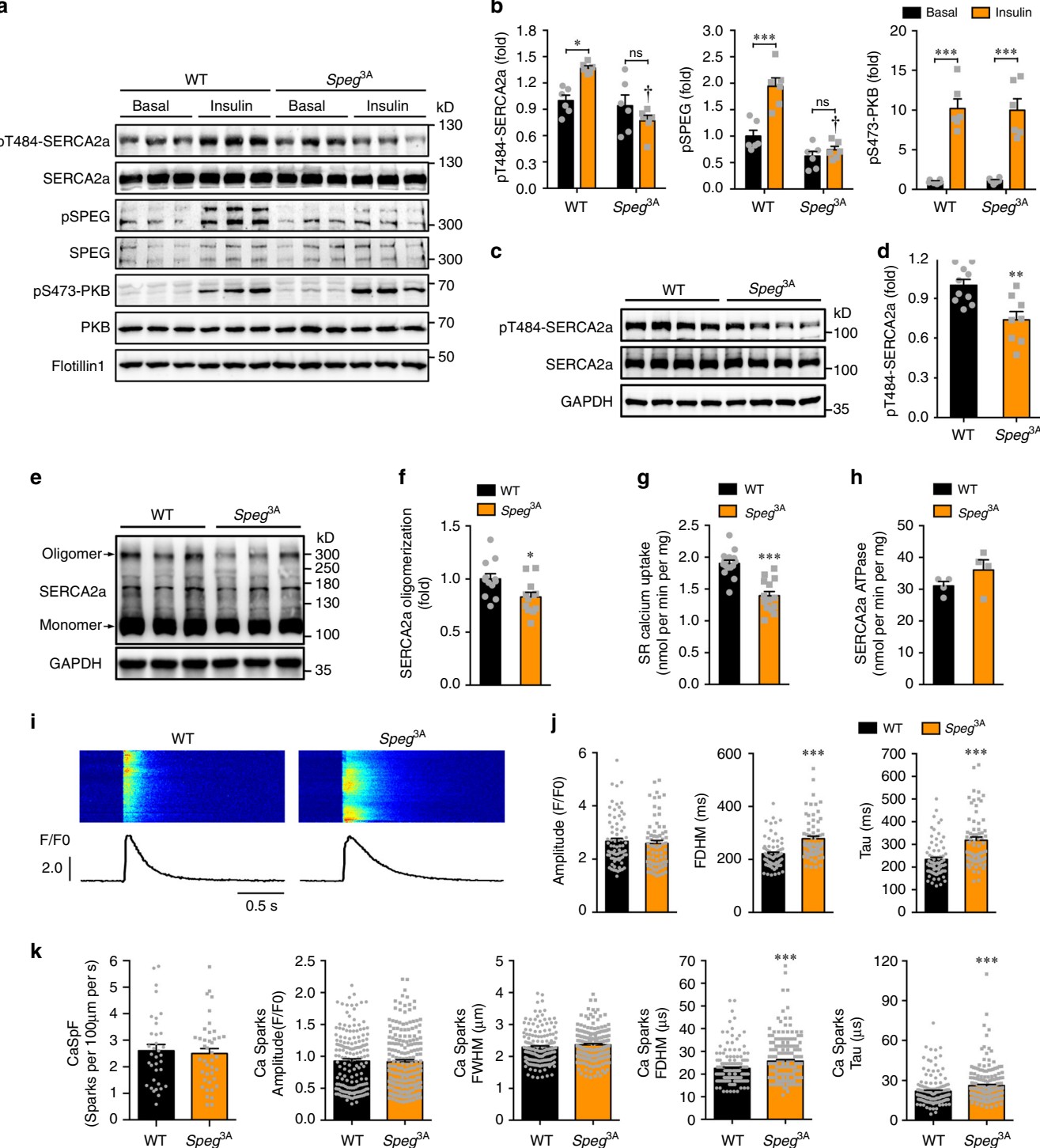

## Discussion

In this study, we show that impaired calcium homeostasis due to cardiac insulin resistance contributes to the development of diabetic cardiomyopathy. We identified SPEG as a PKB substrate in the heart, and found that phosphorylation of SPEG by PKB activated its SK2 but not SK1. We demonstrate that this PKB − SPEG signaling nexus is critical for maintenance of cardiac function through regulating SERCA2a-mediated calcium reuptake into the SR in cardiomyocytes (Fig. 9b). Impairment of this PKB − SPEG signaling nexus may contribute to the development of diabetic cardiomyopathy.

Calcium handling through SERCA2a in cardiomyocytes is impaired in type 2 diabetes, which may be one of the causes for diabetic cardiomyopathy[21,22]. The reasons underlying the impaired SERCA2a activity had not been clear. For instance, its expression levels are either unaltered[21] or slightly decreased in diabetic hearts[22]. Similarly, both unchanged[21] and increased[22] expression of PLB have been observed in diabetic hearts. Moreover, these studies on SERCA2a were performed in the heart from animals with overt diabetes, which displayed insulin resistance as well as metabolic changes. It is therefore difficult to determine whether insulin resistance, or metabolic changes, or both, impairs

**Fig. 8 Calcium homeostasis in *Speg*[3A]-knockin cardiomyocytes. a**, **b** Phosphorylation of PKB, SPEG, and SERCA2a in WT or *Speg*[3A] primary cardiomyocytes stimulated with or without insulin. Phosphorylation of PKB, SPEG and SERCA2a was normalized with their respective total proteins, and quantitative data were shown in **b**. $n = 6$. One-dagger (WT insulin vs *Speg*[3A] insulin) indicates $p < 0.001$. **c**, **d** Thr[484] phosphorylation of SERCA2a in the heart of *Speg*[3A] mice and WT littermates (7-month-old). **c** representative blots. **d** quantitation of SERCA2a-Thr[484] phosphorylation. $n = 10$ (WT) and 8 (*Speg*[3A]). $p = 2.81e-3$. **e**, **f**. Oligomerization of SERCA2a in the heart of *Speg*[3A] mice and WT littermates (7-month-old). **e** representative blots. **f** quantitation data. $n = 12$. $p = 1.97e-2$. G-H. SERCA2a Ca[2+]-transporting activity ($n = 15$ (WT) and 14 (*Speg*[3A]), **g**) and ATPase activity ($n = 4$, **h**) in microsomes isolated from the heart of WT and *Speg*[3A]-knockin mice. $p = 1.98e-6$ (SR calcium uptake) and 0.198 (ATPase activity). **i**, **j** Calcium transients elicited by electrical stimulation in primary cardiomyocytes isolated from the WT and *Speg*[3A]-knockin mice (3-month-old). **i** Representative calcium transient images and curves. **j** Quantitation of amplitudes, full duration at half maximum (FDHM) and time constant Tau of calcium transients. 75 cells from 7 WT mice and 68 cells from 7 *Speg*[3A]-knockin mice were analyzed. $p = 0.603$ (amplitude), $1.14e-6$ (FDHM), and $8.68e-7$ (Tau). **k** Spontaneous calcium sparks in primary cardiomyocytes isolated from the WT and *Speg*[3A]-knockin mice (2-month-old). Frequency, amplitudes, full-width at half maximum (FWHM), full duration at half maximum (FDHM) and time constant Tau of calcium sparks. 168 sparks from 35 cells of four WT mice and 194 sparks from 41 cells of four *Speg*[3A]-knockin mice were analyzed. $p = 0.839$ (frequency), $0.800$ (amplitude), $0.171$ (FWHM), $3.27e-4$ (FDHM), and $5.71e-4$ (Tau). The data are given as the mean ± SEM. Statistical analyses were carried out using two-way ANOVA for **b**, and two-sided *t*-test for **d**, f–**h**, and **j**, **k**. One-asterisk indicates $p < 0.05$, two-asterisk indicates $p < 0.01$, and three-asterisk indicates $p < 0.001$. Source data are provided as a Source Data file.

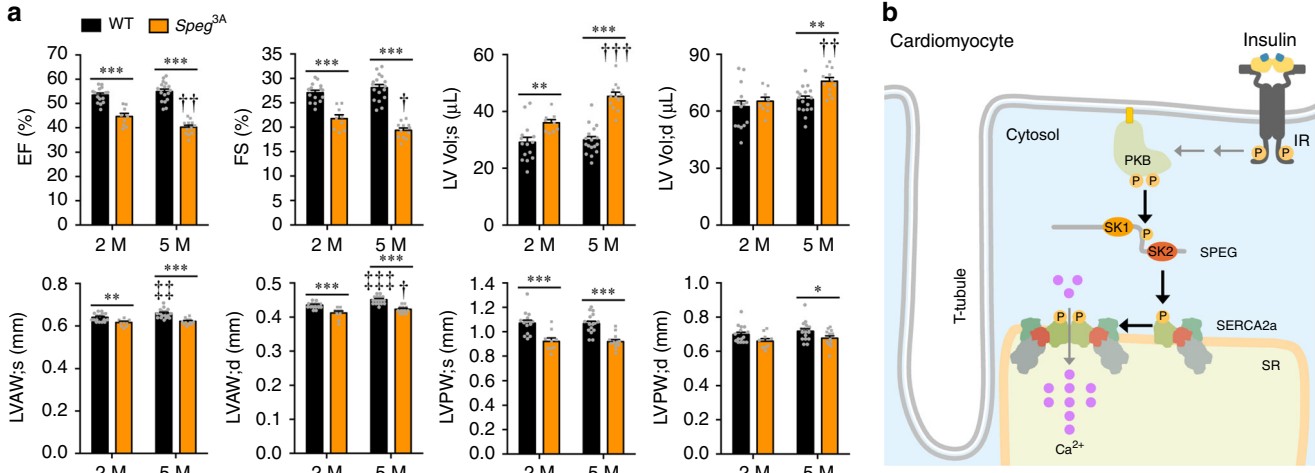

**Fig. 9 Cardiac function of the *Speg*[3A]-knockin mice. a** Echocardiography was performed on the anaesthetized male *Speg*[3A]-knockin mice and wild-type littermates at age of 2 and 5 months to measure EF, FS, LV Vol;s, LV Vol;d, LVAW;s, LVAW;d, LVPW;s, and LVPW;d. $n = 15$ (WT, 2 M), 17 (WT, 5 M), 9 (Speg[3A], 2 M), and 12 (Speg[3A], 5 M). The data are given as the mean ± SEM. Statistical analyses were carried out using two-way ANOVA. One-asterisk (WT vs *Speg*[3A]) and one-dagger (*Speg*[3A] 2 M vs *Speg*[3A] 5 M) indicate $p < 0.05$. Two-asterisk (WT vs *Speg*[3A]), two-dagger (*Speg*[3A] 2 M vs *Speg*[3A] 5 M), and two-diesis (WT 2 M vs WT 5 M) indicate $p < 0.01$. Three-asterisk (WT vs *Speg*[3A]), three-dagger (*Speg*[3A] 2 M vs *Speg*[3A] 5 M), and three-diesis (WT 2 M vs WT 5 M) indicate $p < 0.001$. **b** A diagram represents the proposed model in which the PKB−SPEG signaling nexus links insulin signaling with calcium homeostasis in cardiomyocytes to maintain cardiac function. Upon insulin stimulation, PKB phosphorylates SPEG and activates its second kinase domain, which consequently phosphorylates SERCA2a. Phosphorylation of SERCA2a increases its dimerization that enhances its Ca[2+]-transporting activity. Impaired SPEG phosphorylation by PKB links insulin resistance with cardiac dysfunction through SERCA2a-mediated Ca[2+] reuptake into the SR in cardiomyocytes independent of metabolic assaults. Source data are provided as a Source Data file.

SERCA2a function in such diabetic models. We show here that insulin resistance resulted from PKBα/β deletion or HFD feeding decreases phosphorylation of Thr[484] on SERCA2a. Moreover, our data demonstrate that SPEG phosphorylation by PKB activates its SK2 that in turn phosphorylates SERCA2a. The *Speg*[3A]-knockin mutation prevents insulin-stimulated SPEG phosphorylation and impairs SERCA2a function without causing systemic metabolic changes. Our findings therefore establish a critical role of the PKB-SPEG signaling nexus in regulating SERCA2a function, which involves its Thr[484] phosphorylation by SPEG. These findings may not only help to explain SERCA2a dysfunction in diabetic hearts but also demonstrate a critical role for SPEG in diabetic cardiomyopathy. The risk of hospitalization for heart failure remains high in type 2 diabetic patients who have optimal glycemic control via anti-diabetic drugs[11]. In view of our study here, we suspect that current therapeutic strategies for glycemic control in T2D might not resolve cardiac insulin resistance that may persistently impair calcium homeostasis in cardiomyocytes.

SPEG has recently emerged as a critical regulator for cardiac development and function[12,14–16]. Despite its importance, it remained elusive till now how this protein particularly its kinase activity is regulated in the heart. In part, this was probably due to technical challenges since SPEG is a large protein (~300 kDa) with two kinase domains (SK1 and SK2) in its C-terminal region[14]. Towards understanding its function and regulation, we recently found that the SK1 and SK2 of SPEG may have divergent functions. The SK2 of SPEG can phosphorylate SERCA2a while the SK1 is responsible for phosphorylation of JPH2[16]. Here we show that insulin-stimulated PAS-reactive phosphorylation of SPEG is required for the SK2 activity while it is not a prerequisite for the SK1 activity. This selective activation of SK2 by insulin–PKB signaling might enable these two kinase domains to fulfil their distinct roles in the heart. Besides insulin, other stimuli such as insulin-like growth factor-1 (IGF-1) and angiotensin II (AngII) can activate PKB, which have pro-hypertrophic functions in the heart[23,24]. It is therefore possible that such as

pro-hypertrophic stimuli might also regulate the SPEG-SERCA2a axis through activation of PKB under cardiac hypertrophic conditions. This regulatory axis of PKB-SPEG-SERCA2a might not only link to insulin resistance but also contribute to cardiac dysfunction under other disease conditions. It is still unclear how the SK1 of SPEG is regulated in the heart. It has been reported that SPEG has a putative calmodulin binding site between SK1 and SK2[25], which is close to the PAS-reactive phosphorylation sites identified in this study. SPEG belongs to the MLCK subgroup of CaMK Ser/Thr protein kinase family and its SK1 is more similar to other kinases within the MLCK group than is SK2[14]. Calmodulin binding is a common feature for MLCK kinase members[26] and might regulate the SK1 of SPEG.

Metabolic alterations and dysregulation of calcium homeostasis are two features of the diabetic heart, which interacts with and reinforces each other. The hexosamine biosynthesis pathway integrates metabolism of glucose, lipid and amino acids to produce uridine diphosphate-N-acetylglucosamine for O-linked b-N-acetylglucosamine (O-GlcNAc) modification of proteins[27]. In T2D, the hexosamine biosynthesis pathway is activated, and the resultant O-GlcNAcylation modifies a number of proteins to decrease calcium cycling and sensitivity[28]. On the other hand, changes of cellular calcium may also lead to metabolic alterations. For instance, overexpression of SERCA2a in the heart increases glucose oxidation and simultaneously decreases fatty acid oxidation[29]. Knockout of PLB that enhances SERCA activity results in an increase in oxygen consumption normalized for work[30]. Our data show that the $Speg^{3A}$-knockin mutation does not impair systemic glucose and lipid metabolism. Moreover, it affects neither insulin-stimulated glucose uptake in isolated cardiomyocytes nor expression of key regulators for glucose and lipid metabolism in the heart. However, it is still possible that the decreased SERCA2a activity or impaired cardiac function might impact on cardiac metabolism in the $Speg^{3A}$-knockin mice. Given the importance of SPEG in the heart, a better understanding of whether and how the $Speg^{3A}$-knockin mutation affects cardiac metabolism is needed.

In summary, we show that cardiac insulin resistance impairs calcium homeostasis via the PKB – SPEG – SERCA2a pathway, which contributes to the development of diabetic cardiomyopathy. SPEG may serve as a new target to modulate SERCA2a activation for treatment of diabetic cardiomyopathy.

## Methods

**Materials**. Recombinant human insulin was purchased from Novo Nordisk (Bagsvaerd, Denmark). High fat diet (60 kcal% fat, Cat. No. 12492) was from Research Diets (USA). Protein G-Sepharose was bought from GE-Healthcare (Little Chalfont, Buckinghamshire, UK). Precast NuPAGE® Bis-Tris gels were from Thermo Fisher Scientific (Waltham, MA, USA). Akti1/2 was from Merck Millipore (Darmstadt, Germany), and PI-103 was from Enzo Life Sciences (Farmingdale, NY, USA). All other chemicals were from Sigma-Aldrich (Shanghai, China) or Sangon Biotech (Shanghai, China). The commercial antibodies and resins are listed in Supplementary Data 3. The antibodies recognizing RalGAPα1 and RalGAPα2 were described previously[31], and the pThr484-SERCA2a antibody was as previously reported[16].

**Molecular biology**. The cDNAs encoding mouse SPEG or human SERCA2a were cloned into the vectors pcDNA5-FRT/TO-GFP or pcDNA5-FRT/TO-HA or pcDNA5-FRT/TO-Flag for expression in mammalian cells. Fragmentation and point mutation of SPEG were carried out using standard procedures. The sequence contexts of mutated sites on SPEG are: PGLVRRLsLSLSQKL (Ser2413 in lower case), LAVRRRLsstLERL (Ser2461/Ser2462/Thr2463 in lower case) and FGRLRRAt-sEGESLR (Thr2498/Ser2499 in lower case). SPEG fragments were cloned into the pGEX6P vector for protein expression in E. coli. All DNA constructs were sequenced by Life Technologies (Shanghai, China).

**Generation of the $Speg^{3A}$-knockin mice**. The $Speg^{3A}$-knockin mice on C57Bl/6 J background were generated using the CRISPR/Cas9-based strategy outlined in Fig. 7a by the transgenic facility at Nanjing University. The cluster of serine/threonine residues Ser2461/Ser2462/Thr2463 (the surrounding sequence is

LAVRRRLsstLERL, Ser2461/Ser2462/Thr2463 shown in lower case, and numbering is according to NP_031489.4) on SPEG was substituted to alanine by knockin mutagenesis. An XhoI enzyme restriction site was also introduced by changing the Leu2464 encoding codon CTG to a synonymous codon CTC to facilitate genotyping. The $Speg^{3A}$ mice were genotyped by amplifying the mutated region (562 bp) using two primers (5'-CGGAGGACGACGGCATATAC-3' and 5'-CAGAGCCTGTCTCTAGCACAC-3'), followed by restriction digestion with XhoI (280/282 bp cleaved products for $Speg^{3A}$-knockins). The $Speg^{3A}$-knockin mice were backcrossed to C57Bl/6 J background for at least five generations before experiments.

**Animal breeding and husbandry**. The Ethics Committee at Model Animal Research Center of Nanjing University approved all animal procedures used in this study, which are complied with all relevant ethical regulations. Mice and rats were housed under a light/dark cycle of 12 h, and had free access to food and water unless stated.

PKBαf/f and PKBβ knockout mice were previously reported[32,33], and used to generate mice with PKBα/β double knockout in the heart.

Heterozygote X heterozygote mating was set up to produce $Speg^{3A}$-knockin homozygotes and wild-type (WT) littermates.

**Blood chemistry**. Blood was collected via tail bleeding, and glucose, free fatty acid, triglyceride and total cholesterol in the blood were measured using a Breeze 2 glucometer (Bayer), Wako LabAssay NEFA kit (294-63601), LabAssay Triglyceride (290-63701), and LabAssay Cholesterol kit (294-65801) (Wako Chemicals USA), respectively.

**Insulin injection and oral glucose tolerance test**. Mice were restricted from food access overnight (16 h). For insulin injection, mice were anaesthetised and intra-peritoneally injected with a bolus of insulin (150 mU insulin per g of body weight) for 20 min before terminated by cervical dislocation for tissue collection. For oral glucose tolerance test, mice were administered via oral gavage with a bolus of glucose (1.5 mg glucose per g of body weight). Blood glucose levels were determined at the indicated time points.

**Tissue homogenization and lysis**. Mouse tissues were harvested, snap-frozen in liquid nitrogen, and homogenized in lysis buffer (50 mM Tris-HCl (pH 7.4), 1 mM EDTA, 1 mM EGTA, 1% (v/v) Triton X-100, 1 mM sodium ortho-vanadate, 10 mM sodium glycerophosphate, 50 mM sodium fluoride 5 mM sodium pyrophosphate, 0.27 M sucrose, 2 μM microcystin-LR, 1 mM benzamidine, 0.1% (v/v) 2-mercaptoethanol, 0.2 mM phenylmethanesulfonyl fluoride, 1 mg/ml Leupeptin, 1 mg/ml Pepstatin and 1 mg/ml Aprotinin) using a Polytron homogenizer (Kinematica, Luzern, Switzerland). After lysed on ice for 30 min, tissue homogenates were centrifuged to remove tissue debris. Protein concentrations of tissue lysates were measured using Bradford reagent (Thermo Fisher Scientific).

**Immunoprecipitation and immunoblotting**. For immunoprecipitation of target proteins, tissue or cell lysates were incubated with antibody-coupled protein G-Sepharose or GFP-binder (ChromoTek GmbH, Planegg-Martinsried, Germany) for 16 h at 4 °C. After non-specific binding proteins were removed from resins through washing, immunoprecipitates were eluted in SDS sample buffer.

For immunoblotting, lysates or immunoprecipitates were separated via SDS-PAGE and immunoblotted onto nitrocellulose membranes that were probed with primary and secondary antibodies. Membranes were then incubated with ECL substrates (GE-Healthcare, UK), and chemiluminescence signals were detected using a gel documentation system (Syngene, UK).

**Mass-spectrometry**. Protein immunoprecipitates were electrophoretically separated via SDS-PAGE and stained with Coommassie dye. After excised from gels, protein bands were digested with trypsin. Resultant peptides were further separated via a Dionex 3000 nano liquid chromatography system and analysed by LC-MS on an LTQ-Orbitrap (Thermo Finnigan) mass spectrometer. Mascot generic format (MGF) files were obtained from raw files using raw2msm v1.7 software (Matthias Mann), and searched using a Mascot 2.2 in-house server against the Swiss-Prot database to identify peptides and proteins.

**In vitro phosphorylation**. The recombinant GST-SPEGP2227-S2583 proteins were expressed in E. coli and purified using glutathione-Sepharose 4B (GE-Healthcare). The purified GST-SPEGP2227-S2583 proteins were in vitro phosphorylated by a His-PKB-S473D expressed in insect cells and activated by PDK1 at 30 °C for 30 min. The reaction was stopped by addition of laemmli sample buffer.

**Echocardiography (Echo)**. Mice were anaesthetized with gaseous isoflurane and subjected to Echo analysis via a Vevo 770 high-resolution in vivo micro-imaging system (VisualSonics, inc) with a 30 MHz RMV-707B ultrasonic probe. M-mode pictures were collected and used to determine the following parameters: left ventricle anterior wall (LVAW), left ventricle posterior wall (LVPW), left ventricle

internal dimension (LVID), and left ventricle volume (LV Vol) of systole and diastole. The equation to calculate ejection fraction (EF) is EF% = [(LV Vol;d − LV Vol;s)/LV Vol;d] × 100%, and fractional shortening (FS) is FS% = [(LVID;d − LVID;s)/LVID;d] × 100%.

**Isolation of primary cardiomyocytes**. Primary mouse cardiomyocytes were isolated from the heart of heparin-treated mice using a collagenase-based method [34]. A collagenase solution (1 mg/ml) was perfused into the heart using a Langendorff system (ADInstruments). The resultant cell suspension was filtered through a 100 μm cell strainer. Primary cardiomyocytes were then washed for three times in Krebs-Henseleit buffer B containing 5 mM taurine and 10 mM 2,3-butanedione monoximine with $Ca^{2+}$ (0.1 mM for the first round, 0.2 mM for the second round, and 0.6 mM for the third round).

Primary neonatal rat cardiomyocytes was isolated from ventricles of neonatal animals (postnatal day 0–3, rat strain Sprague Dawley). Minced ventricle cubes were digested with 0.25% trypsin at 4 °C overnight, and further incubated with collagenase (1 mg/ml) at 37 °C for 15 min. After removal of undigested tissue debris, cell suspensions were plated in DMEM containing 10% (v/v) foetal bovine serum for 1 h. Within this period, fibroblasts were settled down and removed. Cardiomyocytes were reseeded in fresh DMEM plus 10% (v/v) foetal bovine serum, and transfected with plasmids using Lipofectamine 3000 reagent (Thermo Fisher Scientific).

**Calcium imaging in primary cardiomyocytes**. Calcium transient assay was carried out in primary rat or mouse cardiomyocytes using a Fluo-4-AM based method[35]. Primary cardiomyocytes were resuspended in Hanks buffer containing 1 mM $MgCl_2$, 1 mM $CaCl_2$ and 2% (w/v) BSA, and incubated with 5 μM Fluo-4-AM (Thermo Fisher Scientific). After incubation, cells were then stimulated using a GRASS S48 stimulator (frequency 0.5 Hz, duration 60 ms, decay 40 ms, voltage 80 V, repeat). A line-scan mode was set for a Zeiss LSM510 confocal microscope to take images that were analyzed using IDL5.5 (Harris Geospatial Solutions). The decay time (Tau) was determined via the period lasting from the peak of calcium transients to 63% from the peak to the basal level in the fading phase.

**Imaging and analysis of t-tubule (TT)**. TT organization was analysed using a Di-8-ANEPPS based method[36]. Primary cardiomyocytes were stained with Di-8-ANEPPS (10 μM) for 15 min. After staining, images were taken using a Carl Zeiss 880 confocal microscope. Fast Fourier Transforms of cell images were used to quantify TT organization. The peak amplitude (called TT power) in the Fourier spectrum of cell images at the TT frequency was analyzed using ImageJ software with a plugin TTorg (http://mirror.imagej.net/plugins/ttorg).

**Measurements of $Ca^{2+}$-ATPase activity and $Ca^{2+}$ uptake**. The ATPase activity of SERCA2 was determined in microsomes containing crude SR membrane vesicles via measurement of inorganic phosphate (Pi) released from ATP hydrolysis[37]. The reaction was carried out by incubating microsomes (50 μg protein) with an assay buffer containing 100 mM KCl, 10 mM HEPES (pH 7.4), 5 mM $MgCl_2$, 100 μM $CaCl_2$, 1.5 mM ATP, 2 μM A23187, and 5 mM sodium azide in the absence (total activity) or presence of 5 μM thapsigargin (activity of thapsigargin-insensitive calcium pumps) at 30 °C for 30 min, and stopped by adding ice-cold 10% TCA. The thapsigargin-insensitive $Ca^{2+}$-ATPase activity was subtracted from total activity to obtain the activity of thapsigargin-sensitive $Ca^{2+}$-ATPase (SERCA2-ATPase).

$Ca^{2+}$ uptake in microsomes was determined using a Fura-2 based method[38]. Isolated microsomes were incubated with 2 μM Fura-2 free acid in assay buffer (100 mM KCl, 10 mM HEPES-KOH (pH 7.4), 10 mM oxalate, 5 mM $MgCl_2$, and 10 μM ruthenium red). The uptake of Fura-2 into microsomes was initiated by addition of 5 mM ATP and 2 μM $Ca^{2+}$. The fluorescence was excited at 340 and 380 nM, respectively, and recorded at 510 nM emission using a fluorescence microplate reader (BioTek). Free $Ca^{2+}$ in the microplate was calculated from the Fura-2 fluorescence using the equation, Free calcium = $Kd \times \beta \times \frac{(R-R_{min})}{(R_{max}-R)}$, where R is the ratio of 510-nm emission fluorescence intensity excited at 340 and 380 nM[39]. Rmax and Rmin were determined by addition of 10 mM $Ca^{2+}$ or 25 mM EGTA in assay buffer, respectively. The assumed dissociation constant (Kd) for Fura-2/$Ca^{2+}$ was 200 nM[39]. β is the ratio of fluorescence intensity of $Ca^{2+}$ free and $Ca^{2+}$ bound form of Fura-2 at 380 nM. The free $Ca^{2+}$ versus assay duration was analysed using Clampfit 10.4 (Molecular Devices). The linear portion of the slope after addition of $Ca^{2+}$ was used for calculation of $Ca^{2+}$ uptake rates in microsomes.

**Cell culture and transfection**. Human embryonic kidney HEK293 cells and rat H9C2 cardiomyocytes were obtained from the Cell Resource Center, Chinese Academy of Medical Sciences and Peking Union Medical College (China), and cultured in DMEM medium containing 10% (v/v) foetal bovine serum. Tests for mycoplasma contamination were carried out regularly. Transfection of cells with plasmid DNA was carried out using Lipofectamine 3000 reagent (Thermo Fisher Scientific). Cells were lysed in lysis buffer on ice for 30 min at 2 days after transfection. H9C2 cardiomyocytes were differentiated in DMEM medium containing 1% (v/v) foetal bovine serum and 1 μM retinoic acid for 7 days.

**Calcium transient assay in HEK293 cells**. HEK293 cells expressing SERCA2a together with SPEG or an empty vector were incubated with 5 μM Fluo-4-AM. Afterwards, cells were stimulated with 200 μM carbamylcholine. A frame scan mode was set for an Olympus confocal microscope to take images of cells for ~270 sec.

**Fluorescence resonance energy transfer (FRET) assay**. ECFP-SERCA2a and EYFP-SERCA2a plasmids (1:1 molar ratio) were transfected into HEK293 cells together with HA-SPEG or empty vector. Two days after transfection, cells co-expressing ECFP-SERCA2a and EYFP-SERCA2a were selected for FRET assay using a Leica SP5 confocal microscope, as described in the Leica FRET Sensitized Emission application manual. Calculation of FRET efficiency was performed using the following formula: FRET Efficiency (%) = (FRET signal-β*Donor Signal-γ*Acceptor signal)/(Acceptor signal), where β is obtained with donor only specimen and calculated as β = $Signal_{indirectAcceptor}/Signal_{Donor}$, and γ is obtained with acceptor only specimen and calculated as γ = $Signal_{IndirectAcceptor}/Signal_{DirectAcceptor}$.

**Statistical and reproducibility**. Data were analyzed via t-test for two groups, or via one-way or two-way ANOVA for multiple groups using Prism software (GraphPad, San Diego, CA, USA), and differences were considered statistically significant at $p < 0.05$. Individual data points are shown when $n \le 10$.

Except for mass-spec experiments, similar results were obtained from at least two experiments.

**Reporting summary**. Further information on research design is available in the Nature Research Reporting Summary linked to this article.

## Data availability
All data generated or analysed during this study are included in this published article (and its supplementary information files). The proteomic datasets can be downloaded from the MassIVE Repository in University of California, San Diego (ftp://massive.ucsd.edu/MSV000085228/ and ftp://massive.ucsd.edu/MSV000085229/). All remaining data are available from the corresponding author upon reasonable request.

The source data underlying Figs. 1a-f, 2a-f, 3a-g, 4a-e, 4g-k, 5a-d, 6a-e, 7b-d, 8a-h, 8j-k and 9a and Supplementary Figs 1a-l, 2, 3b, c, f, 4a-g, 5a, 6b-c, 7b, 8a-g, 9a-d, 10a, c, 11, 12a and d are provided as a Source Data file.

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

## Acknowledgements

We thank members of the resource unit at Nanjing University for technical assistance. Thanks to the Ministry of Science and Technology of China (Grant Nos. 2018YFA0801102 to S.C. and 2018YFA0801104 to H.Y.W.), the National Natural Science Foundation of China (Grant Nos. 31671456 and 31971067 to H.Y.W.), and the Science and Technology Foundation of Jiangsu Province of China (Grant Nos. BK20161393 (Basic Research Program) to H.Y.W., and BK20190305 (Basic Research Program) to Q.L.C.), for financial support.

## Author contributions

C.Q., Q.D., M.L., R.Z.W., Q.O.Y., S.S., S.S.Z., Q.L.C., Y.S., L.C., H.W., D.G.C., and K.F.O.Y. performed experiments, analyzed data and reviewed the manuscript. Z.Z.Y. reviewed the manuscript. C.M. reviewed and edited the manuscript. H.Y.W. and S.C. designed experiments, analyzed data, and wrote the manuscript. S.C. is the guarantor of this study. All authors approved the final version of the manuscript.

## Competing interests

The authors declare no competing interests.
