## [Peer Review File · Nature Communications]

Reviewers' comments:

Reviewer #1 (Remarks to the Author):

The authors of the present manuscript nicely proved the role of SPEG in the regulation of SERCA activity downstream insulin. By using several elegant mouse and cellular models, they mainly showed that (i) SPEG is phosphorylated on Ser2461/Ser2462/Thr2463 by PKB in response to insulin; (ii) PKB-mediated phosphorylation of SPEG activates its SK2 domain, which in turn phosphorylates SERCA2a, accelerating calcium re-uptake into the SR; (iii) A mouse model bearing a Speg3A knockin mutation displayed cardiac dysfunction; (iv) Speg3A knockin mutation impairs SERCA2a phosphorylation and calcium re-uptake into the SR.

They conclude that phosphorylation of SPEG by PKB activates its SK2 and the PKB-SPEG signalling nexus is important for maintenance of cardiac function through regulating SERCA2a-mediated calcium re-uptake into the SR in cardiomyocytes.

In a global view, this manuscript is composed of solid data obtained both in vitro and in vivo. The authors found a novel mechanism involved in cardiac function maintenance and that could be potentially disrupted in important cardiac diseases. However, I have several concerns.

Major concerns:

- My first criticism comes from the control mice used in figure 1. The control use is mice without MCM (with/without Tam?). However, MCM expression with tamoxifen are known to induce transient cardiac dysfunction (Molkentin publication). Here, dysfunction found in PKB α ff;MCM;PKB β -/- looks clearly more dramatic; however, it is possible that this dramatic dysfunction is due to an excessive negative impact of active MCM expression in combination to PKB β -/- deletion. The authors should show that MCM/tamoxifen do not provoke any phenotype in PKB β -/- mice.

- Figure 1A shows only partial deletion of PKB α in heart extracts. Is it due to the expression in non-cardiomyocytic cells or due to non-maximal MCM/tam efficacy? As the authors purified cardiomyocytes from these hearts, a simple WB in purified cells would be able to answer to this question.

- Related to figure 1: Characterization of the dysfunction found in these mouse hearts should go deeper. What happens in these hearts? Is there any cardiomyocyte death? Is fibrosis occurring? Is there any cardiomyocyte hypertrophy? all the echocardiographic parameters should be shown (in table for example).

- Figure 2C-D: the group PKB KO under basal condition is lacking. This should be shown. What is the impact of PKB KO on basal signalling? This is important to state that these hearts are “insulin resistant”
- Figure 2E: The reviewer does not understand why the authors did not choose to use PKB inhibitor rather than PI3K inhibitor? The use of a direct PKB inhibitor would be a better proof of the role of PKB. Moreover, the blots presented are not totally convincing (total GFP-SPEG looks similarly increased than PAS band in insulin treatment).
- Figure 7: similar to point 2, the characterization of the dysfunction found in these mouse hearts should go deeper. What happens in these hearts? Is there any cardiomyocyte death? Is fibrosis occurring? Is there any cardiomyocyte hypertrophy?
- All along the manuscript, the authors linked PKB-SPEG-SERCA to insulin action. However, PKB is also targeted by other stimuli such as IGF-1 (under basal and hypertrophic conditions) and pro-hypertrophic molecules (AngII, ...). In other words, the new paradigm revealed by the authors could also play a role in other situation than insulin resistance/diabetes. The reviewer is pretty convinced that such paradigm should play a role in cardiac dysfunction following TAC surgery. All these aspects should be highlighted.
- The authors start their discussion by affirming that “they show that impaired calcium homeostasis due to cardiac insulin resistance contributes to the development of diabetic cardiomyopathy independent of metabolic assaults. ». However, they never evaluate the role of SPEG-induced SERCA phosphorylation in a model of diabetic cardiomyopathy. They showed that SPEG mutant KI provokes SERCA and cardiac dysfunction. If they want to conclude on diabetic cardiomyopathy, they should put their KI mice in HF/HS diet.
- In their abstract, the authors stated that they showed that “A mouse model bearing a Speg3A KI mutation displayed cardiac dysfunction without overt metabolic changes ». However, the authors never evaluated the metabolic flexibility of the heart of these KI mice. Are they characterized by normal glucose/FA uptake and oxidation? The authors only showed that there are no apparent global plasma parameter changes
- The reviewer does not understand why the authors did not want to see if their paradigm is compatible with what really happens in diabetic cardiomyopathy? It would be so simple to see if SPEG phosphorylation is affected in model of HF/HS diet animals and if this correlates with disruption of SERCA2a phosphorylation, with Ca dysfunction and cardiac dysfunction. If yes, a time course of events would confirm the possible implication of SPEG alteration in other events.
- The title includes “diabetic cardiomyopathy”. However, this pathology was not really studied in the present manuscript.
- All along the manuscript, the great majority of blots are presented without quantification and statistical analysis. This should be done.

Reviewer #2 (Remarks to the Author):

The authors have extended their recent seminal study (Circ Res 2018) in which they showed that SPEG could phosphorylate SERCA2a and enhance SR Ca-ATPase function in cardiac myocytes. They do add some additional mechanistic depth to that former study, but they falter rather badly in making a compelling case for the criticality of this PKB-SPEG-SERCA2a pathway in being a critical link in diabetic cardiomyopathy as implied in the title abstract and discussion. There are many strengths in this work, but there are a number of areas that reflect weaknesses in either experimental design, analysis or interpretation.

1. The data show a slowing of fluorescence decay (Ca transient decline) in several cases. In Fig 1F for PKB^{Δ/Δ} KO, this is interpreted as due to slowed SR Ca uptake and taken to be causally responsible for the contractile deficit observed in echos (Fig 1B-C). Several issues arise.

First, these tau values for [Ca]_i decline in mouse ventricular myocytes are quite slow and might suggest overloading with Ca indicator (and buffering of Ca transients) which slows the kinetics of Ca transients (and the extent of indicator overloading could be responsible for the differences seen).

Second, the amplitudes of Ca transients are not reported, so we really have no idea whether the changes in Ca handling are at all related to the functional defects.

Third, based on their prior study (and later data here in other myocytes, Fig 6) these changes in [Ca]_i decline kinetics occurred despite unaltered Ca transient amplitudes. If the amplitude is unchanged and the kinetics of [Ca]_i decline are slowed this would promote a stronger active state, and be the opposite of the reduced EF and FS.

Fourth, while this leaves the door open for a potential myofilament associated defect in contractility caused by the PKB KO (Fig 1) or SPEG-3A mice, that notion is not addressed at all (and would be opposed to their central conclusion).

Fifth, there are no measurements at all of either SR Ca content or SR Ca leak. Without that sort of data, it is hard to interpret what is really going on with myocyte Ca handling. For example, if reduced SR Ca uptake is responsible for the slowed [Ca]_i decline in PKB-KO (Fig 1), SPEG-3A KI (Fig 6) and SPEG D3098A (Fig 4), then that ought to reduce SR Ca content and reduce Ca transient amplitude. But in the one case where Ca transient amplitudes are reported, they are unchanged. That needs mechanistic explanation. Ca spark decline rate is primarily a function of diffusion and RyR properties and SERCA2 function is a minor contributor. Thus, without data on SR Ca load or leak (via RyR), the spark data is also difficult to assign mechanistic interpretation.

Sixth, phospholamban phosphorylation is the main acute regulator of SERCA2a function in adult ventricular myocytes, and this seems essentially unconsidered or explored.

2. The SR Ca-ATPase is known to be an energetically highly efficient, with nearly 2 Ca transported for each ATP used, and able to create a [Ca] gradient across the SR membrane that approaches the thermodynamic limit. Claiming a major increase in Ca transport efficiency (more Ca transported per ATP) is an issue. Moreover, their prior study implied coupling ratios above 100 Ca transported per ATP split (which thermodynamically is impossible). This gives one concern as to the precise mechanism by which phosphorylation of SERCA2a at T484 enhances SERCA2 function. This was not especially well addressed in the prior study or here.

3. PKB has also been shown to enhance SR Ca leak via CaMKII-dependent phosphorylation of RyR2 (PMID: 28476660). Conceivably there is more than one thing going on, and increased SR Ca leak combined with enhanced SR Ca uptake could result in unaltered SR Ca load and Ca transient amplitudes. That is not probed.

4. There are numerous cases where controls are missing entirely (e.g. why isn't PKB-KO shown at baseline without insulin in Fig 2C). There are several other examples of this sort of oversight. It is stated that PKB deficiency causes insulin resistance in mouse (pg 5), but that is neither shown nor referenced.

5. The link of this PKB-SPEG-SERCA pathway in diabetic cardiomyopathy is lost in the shuffle. In many overexpression studies in HEK cells (and even in myocytes) it is unclear that diabetes or insulin activation are involved in the measurements (e.g. it does not look like insulin was required to activate PKB/SPEG effects in Fig 4). Also many gels are neither quantified or statistically analyzed (e.g. Figs 2-4).

Reviewer #3 (Remarks to the Author):

This is an interesting manuscript by Quan and colleagues potentially linking PKB-SPEG signaling with insulin resistance and the subsequent consequences of abnormal calcium homeostasis in cardiomyocytes during diabetic cardiomyopathy. The experiments were elegantly performed, and the results are convincing using genetically modified mouse models of disease. Insulin resistance in the mouse heart was shown using an inducible PKB α/β deficient mouse. Moreover, a Speg3A knockin mutant mouse supported the in vivo function of Speg phosphorylation by insulin.

Mutation of the Speg gene has been shown previously to lead to a dilated cardiomyopathy in the developing heart and death shortly after birth. Speg mutations have also been associated with a dilated cardiomyopathy in human patients. Moreover, inducible disruption of Speg in the heart has shown altered calcium homeostasis and cardiac dysfunction. The present manuscript advances our understanding and provides new insight into the physiologic function of Speg, with a focus on the abnormal consequences of diabetic cardiomyopathy and abnormal calcium homeostasis.

While the authors make interesting points about the ability to promote insulin resistance in the hearts without other metabolic changes, I believe it is still critical to demonstrate that in a rodent model of diabetes or obesity-induced insulin resistance (and not a genetically modified mouse) that there is evidence of altered Speg phosphorylation in the proposed serine/threonine residues. If this does occur, is there altered calcium homeostasis (including decreased phosphorylation of Thr484 on SERCA2a) and associated cardiomyopathy. Using a pathophysiologic model of diabetes or obesity-induced insulin resistance, even in the presence of metabolic abnormalities, will provide additional support to the underlying conclusion of the manuscript regarding Speg, insulin resistance, and diabetic cardiomyopathy.

The description of the statistical analyses appears to be appropriate, however providing further information about the specific statistical analysis performed for each figure in the legend would help the readers to better understand the analysis.

Reviewer #4 (Remarks to the Author):

Report on manuscript submitted to Nature Communications titled:

"A PKB-SPEG signaling nexus links insulin resistance with diabetic cardiomyopathy by regulating calcium homeostasis"

Authors: Quan C et al from the research group of Prof S Chen

The authors delineated the signalling events leading from the activation of PKB/Akt to SERCA2a phosphorylation and the function of re-uptake of calcium by the SR. To do so, they utilized biochemical techniques and a mouse model with a specific knockin mutation of the protein of interest, SPEG. This research was built on their previous work showing that SPEG controls calcium re-uptake into the sarcoplasmic reticulum.

The investigation was well planned and formed a cohesive set of evidence to prove the interactions as described. The research generated an enormous amount of very interesting results. The evidence supplied in the paper substantiates the conclusion. The findings are novel and will be of interest to a

wider spectrum of researchers, both from the field of insulin resistance as well as cardiac contractility.

I do have however, have a broad comment that the model utilized simulated a metabolic defect that will arise in a state of insulin resistance – as also stated in the rationale of the research project. Despite the fact that the impairments were observed without metabolic derangements in the animals, to conclude that “a diabetic cardiomyopathy” can develop independent of metabolic insults, is not completely correct. Can the authors list any other insult that will allow the SPEG protein to not be phosphorylated by PKB? If not, they have actually demonstrated an end-point of the result of metabolic derangements and should change their conclusion as such.

It is stated in the methods that different statistical analyses were performed on data sets but it should be listed in the Figure legends, together with the n-values, what analysis method was performed to analyse that specific set of results.

Minor points:

On page 10, second paragraph line 5: remove one of the “in” word.

Page 35, supplementary table 2: In the legend it is stated that the work was performed using IGF1 as ligand but the table states Insulin + or -. Please correct.

Reference 16 is incomplete

Ref: Ms. NCOMMS-19-07467A

Title: A PKB-SPEG signaling nexus links insulin resistance with diabetic cardiomyopathy by regulating calcium homeostasis

Thanks a lot to the reviewers for the positive and helpful reviews of our paper. We had already considered some of the points raised, which meant that some of the requested data had already been collected, while other experiments were set up in response to the review letter. We have obtained substantial amounts of new data to strengthen our study and conclusions. Here we outline our responses to comments from the reviewers.

Reviewers' comments:

Reviewer #1 (Remarks to the Author):

The authors of the present manuscript nicely proved the role of SPEG in the regulation of SERCA activity downstream insulin. By using several elegant mouse and cellular models, they mainly showed that (i) SPEG is phosphorylated on Ser2461/Ser2462/Thr2463 by PKB in response to insulin ; (ii) PKB-mediated phosphorylation of SPEG activates its SK2 domain, which in turn phosphorylates SERCA2a, accelerating calcium re-uptake into the SR ; (iii) A mouse model bearing a Speg3A knockin mutation displayed cardiac dysfunction; (iv) Speg3A knockin mutation impairs SERCA2a phosphorylation and calcium re-uptake into the SR.

They conclude that phosphorylation of SPEG by PKB activates its SK2 and the PKB-SPEG signalling nexus is important for maintenance of cardiac function through regulating SERCA2a-mediated calcium re-uptake into the SR in cardiomyocytes.

In a global view, this manuscript is composed of solid data obtained both in vitro and in vivo. The authors found a novel mechanism involved in cardiac function maintenance and that could be potentially disrupted in important cardiac diseases.

- We thank this reviewer for the constructive comments and highly appreciate that he/she considers that our study 'found a novel mechanism involved in cardiac function maintenance and that could be potentially disrupted in important cardiac diseases'. In the revised manuscript, we provide new evidence to strengthen our conclusions.

However, I have several concerns.

Major concerns:

1. My first criticism comes from the control mice used in figure 1. The control use is mice without MCM (with/without Tam?). However, MCM expression with tamoxifen are known to induce transient cardiac dysfunction (Molkentin publication). Here, dysfunction found in PKB α ^{f/f};MCM;PKB β ^{-/-} looks clearly more dramatic; however, it is possible that this dramatic dysfunction is due to an excessive negative impact of active MCM expression in combination to PKB β ^{-/-} deletion. The authors should show that MCM/tamoxifen do not provoke any phenotype in PKB β ^{-/-} mice.

- The control PKB α ^{f/f};PKB β ^{-/-} mice in Figure 1 were also treated with tamoxifen. We performed experiments as suggested, and show that MCM/tamoxifen did not provoke cardiac dysfunction in PKB β ^{-/-} mice (new Supple. Fig. 1M in the revised manuscript).

2. Figure 1A shows only partial deletion of PKB α in heart extracts. Is it due to the expression in non-cardiomyocytic cells or due to non-maximal MCM/tam efficacy? As the authors purified cardiomyocytes from these hearts, a simple WB in purified cells would be able to answer to this question.

- We isolated cardiomyocytes from PKB α^{ff} ;MCM;PKB $\beta^{-/-}$ mice, and could see that PKB α was greatly decreased in PKB α^{ff} ;MCM;PKB $\beta^{-/-}$ cardiomyocytes as compared to that in PKB α^{ff} ;PKB $\beta^{-/-}$ cardiomyocytes after tamoxifen induction (new Supple. Fig. 1B-C in the revised manuscript). However, some residual signals for PKB α could still be detected in PKB α^{ff} ;MCM;PKB $\beta^{-/-}$ cardiomyocytes (new Supple. Fig. 1B-C in the revised manuscript). Together, these data suggest that both expression of PKB α in non-cardiomyocytic cells and non-maximal MCM/tam efficacy results in residual PKB α signals in heart extracts.

3. Related to figure 1: Characterization of the dysfunction found in these mouse hearts should go deeper. What happens in these hearts? Is there any cardiomyocyte death? Is fibrosis occurring? Is there any cardiomyocyte hypertrophy? all the echocardiographic parameters should be shown (in table for example).

- We now show the rest of echocardiographic parameters in the new Supple. Fig. 1G-L of the revised manuscript. The hearts of PKB α^{ff} ;MCM;PKB $\beta^{-/-}$ mice weighed similarly to those of PKB α^{ff} ;PKB $\beta^{-/-}$ mice (new Supple. Fig. 1F). The PKB α^{ff} ;MCM;PKB $\beta^{-/-}$ cardiomyocytes exhibited cell sizes similar to the PKB α^{ff} ;PKB $\beta^{-/-}$ cardiomyocytes (new Supple. Fig. 1E). TUNEL staining and expression of apoptotic markers showed an increase of cardiomyocyte death in the hearts of PKB α^{ff} ;MCM;PKB $\beta^{-/-}$ mice (new Supple. Fig. 2C-D). Expression of Col1A1 was significantly increased and masson's staining showed that fibrosis occurred in the hearts of PKB α^{ff} ;MCM;PKB $\beta^{-/-}$ mice (new Supple. Fig. 2B and 2E). Moreover, markers of heart failure such as ANP, BNP and β MHC were significantly up-regulated in the hearts of PKB α^{ff} ;MCM;PKB $\beta^{-/-}$ mice (new Supple. Fig. 2B).

4. Figure 2C-D: the group PKB KO under basal condition is lacking. This should be shown. What is the impact of PKB KO on basal signalling? This is important to state that these hearts are "insulin resistant"

- We analyzed the PKB α/β KO hearts under basal as well as insulin conditions, and found that PAS-reactive phosphorylation of SPEG and phosphorylation of SERCA2a were decreased under both conditions (new Supple. Fig. 3D, new Fig. 5A-B). The insulin response was suppressed in the PKB α/β KO cardiomyocytes, showing that they developed insulin resistance (new Fig. 5A-B in the revised manuscript). Furthermore, we found insulin increased SERCA2a-Thr⁴⁸⁴ phosphorylation in the WT cardiomyocytes (new Fig. 5A-B in the revised manuscript). In contrast, insulin did not increase SERCA2a-Thr484 phosphorylation in the PKB α/β KO cardiomyocytes (new Fig. 5A-B in the revised manuscript).

5 Figure 2E: The reviewer does not understand why the authors did not choose to use PKB inhibitor rather than PI3K inhibitor? The use of a direct PKB inhibitor would be a better proof of the role of PKB. Moreover, the blots presented are not totally convincing (total GFP-SPEG looks similarly increased than PAS band in insulin treatment).

- We agree with the reviewer and included a direct PKB inhibitor Akti1/2 alongside the PI3K inhibitor PI-103 to treat cells. Insulin stimulated PAS-reactive phosphorylation of

SPEG, which was prevented by pre-treatment with Akt1/2 and PI-103 (new Fig. 2E in the revised manuscript). The quantification data was shown in the new Fig. 2F in the revised manuscript.

6. Figure 7: similar to point 2, the characterization of the dysfunction found in these mouse hearts should go deeper. What happens in these hearts? Is there any cardiomyocyte death? Is fibrosis occurring? Is there any cardiomyocyte hypertrophy?

- We found that the *Speg*^{3A} knockin cardiomyocytes exhibited cell sizes similar to the WT control cardiomyocytes (new Supple. Fig. 8A). No increased cardiomyocyte death was observed in the hearts of *Speg*^{3A} knockin mice as shown by TUNEL staining (new Supple. Fig. 11C-D). Masson's staining showed that no obvious fibrosis occurred in the hearts of *Speg*^{3A} knockin mice (new Supple. Fig. 11B). Moreover, expression of ANP, BNP and β MHC were normal in the hearts of *Speg*^{3A} knockin mice (new Supple. Fig. 11A). These data show that the *Speg*^{3A} knockin mutation causes no apparent cardiac remodeling within the experimental period although it impairs cardiac function.

7. All along the manuscript, the authors linked PKB-SPEG-SERCA to insulin action. However, PKB is also targeted by other stimuli such as IGF-1 (under basal and hypertrophic conditions) and pro-hypertrophic molecules (AngII, ...). In other words, the new paradigm revealed by the authors could also play a role in other situation than insulin resistance/diabetes. The reviewer is pretty convinced that such paradigm should play a role in cardiac dysfunction following TAC surgery. All these aspects should be highlighted.

- We thank this reviewer for pointing this out, and agree that the PKB-SPEG-SERCA2a pathway identified in this study may be targeted not only by insulin but also by other stimuli such as IGF-1 and AngII. Such paradigm may also play a role in cardiac dysfunction following TAC surgery, which is worth of a thorough investigation in future. We include a discussion on these aspects in the revised manuscript (Page 12 line 22-27 in the revised manuscript).

8 The authors start their discussion by affirming that "they show that impaired calcium homeostasis due to cardiac insulin resistance contributes to the development of diabetic cardiomyopathy independent of metabolic assaults. ». However, they never evaluate the role of SPEG-induced SERCA phosphorylation in a model of diabetic cardiomyopathy. They showed that SPEG mutant KI provokes SERCA and cardiac dysfunction. If they want to conclude on diabetic cardiomyopathy, they should put their KI mice in HF/HS diet.

- We first utilized a cardiomyocyte model to study impacts of insulin resistance on phosphorylation of SPEG and SERCA2a. Insulin resistance induced by chronic insulin treatment in H9C2 cardiomyocytes expectedly impaired insulin-stimulated PKB phosphorylation. Importantly, insulin resistance in H9C2 cardiomyocytes inhibited insulin-stimulated phosphorylation of SPEG and SERCA2a (new Fig. 5C-D). Diet-induced obese (DIO) causes insulin resistance in the heart and impairs cardiac function (new Fig. 6). We then utilized hearts from DIO mice as a model of diabetic cardiomyopathy and examined phosphorylation of SPEG and SERCA2a in this model. Insulin-stimulated phosphorylation of SPEG and SERCA2a was blunted in the cardiomyocytes of DIO mice (new Fig. 6). Moreover, the FDHM and Tau of calcium transients were increased and their peaks were decreased in the cardiomyocytes of DIO mice (new Fig. 6).

9. In their abstract, the authors stated that they showed that “A mouse model bearing a *Speg3A* KI mutation displayed cardiac dysfunction without overt metabolic changes ». However, the authors never evaluated the metabolic flexibility of the heart of these KI mice. Are they characterize by normal glucose/FA uptake and oxidation? The authors only showed that there are no apparent global plasma parameter changes.

- We agree with this reviewer, and have changed the sentence to “A genetically-modified mouse model bearing a *Speg*^{3A} knockin mutation to prevent its phosphorylation by PKB displayed cardiac dysfunction without systemic metabolic changes”. We analyzed expression of key factors for glucose and lipid metabolism in the heart, and found no apparent change in their expression in the *Speg*^{3A} knockin heart (new Supple. Fig. 7D-G). We also performed glucose uptake assay in primary cardiomyocytes isolated from the *Speg*^{3A} knockin and WT mice, and found no difference between the two genotypes (new Supple. Fig. 7C). It has been reported that changes in calcium homeostasis due to overexpression of SERCA2a in the heart result in an increase of glucose oxidation and a simultaneous decrease of fatty acid oxidation (PMID: 17142343). Therefore, it is possible that the *Speg*^{3A} knockin mutation might impact on cardiac metabolism via inhibition of SERCA2a activity or via other unknown targets. We have generated a SERCA2a^{T484A} knockin mouse model. Our preliminary data show that the SERCA2a^{T484A} knockin mutation impairs cardiac function (Letter Fig. 1). These two knockin models shall allow us to study possible effect of the SPEG-SERCA2a pathway on cardiac metabolism. To achieve this goal, we need to set up reliable systems such as working heart, ³¹P-NMR spectroscopy and *in vivo* metabolic flux assay, which we don't have yet in our lab. Nevertheless, we add the new data in the revised manuscript and also include a discussion on the possible effect of *Speg*^{3A} knockin mutation on cardiac metabolism (Page 13 line 4-21 in the revised manuscript).

10. The reviewer does not understand why the authors did not want to see if their paradigm is compatible with what really happens in diabetic cardiomyopathy? It would be so simple to see if SPEG phosphorylation is affected in model of HF/HS diet animals and if this correlates with disruption of SERCA2a phosphorylation, with Ca dysfunction and cardiac dysfunction. If yes, I time course of events would confirm the possible implication of SPEG alteration in other events.

- We utilized hearts from DIO mice as a model of diabetic cardiomyopathy and examined phosphorylation of SPEG and SERCA2a in this model. Cardiac function was impaired in DIO mice (new Fig. 6). Insulin-induced phosphorylation of SPEG and SERCA2a was indeed impaired in cardiomyocytes from these mice (new Fig. 6). Moreover, the FDHM and Tau of calcium transients were increased and their peaks were decreased in the cardiomyocytes of DIO mice (new Fig. 6).

11. The title includes “diabetic cardiomyopathy”. However, this pathology was not really studied in the present manuscript.

- As mentioned above, we utilized hearts from DIO mice as a model of diabetic cardiomyopathy in the revised manuscript.

12. All along the manuscript, the great majority of blots are presented without quantification and statistical analysis. This should be done.

- As suggested, we quantified the blots as many as we can throughout the manuscript, and performed statistical analysis.

Reviewer #2 (Remarks to the Author):

The authors have extended their recent seminal study (Circ Res 2018) in which they showed that SPEG could phosphorylate SERCA2a and enhance SR Ca-ATPase function in cardiac myocytes. They do add some additional mechanistic depth to that former study, but they falter rather badly in making a compelling case for the criticality of this PKB-SPEG-SERCA2a pathway in being a critical link in diabetic cardiomyopathy as implied in the title abstract and discussion. There are many strengths in this work, but there are a number of areas that reflect weaknesses in either experimental design, analysis or interpretation.

- We thank this reviewer for finding 'many strengths in this work', and also appreciate his/her comments on 'areas that reflect weaknesses'. We performed a number of new experiments during the revision, and also carefully re-analyzed our data and pertinently interpreted them, which strengthen our study.

1. The data show a slowing of fluorescence decay (Ca transient decline) in several cases. In Fig 1F for PKB / KO, this is interpreted as due to slowed SR Ca uptake and taken to be causally responsible for the contractile deficit observed in echos (Fig 1B-C). Several issues arise.

First, these tau values for [Ca]_i decline in mouse ventricular myocytes are quite slow and might suggest overloading with Ca indicator (and buffering of Ca transients) which slows the kinetics of Ca transients (and the extent of indicator overloading could be responsible for the differences seen).

Second, the amplitudes of Ca transients are not reported, so we really have no idea whether the changes in Ca handling are at all related to the functional defects.

Third, based on their prior study (and later data here in other myocytes, Fig 6) these changes in [Ca]_i decline kinetics occurred despite unaltered Ca transient amplitudes. If the amplitude is unchanged and the kinetics of [Ca]_i decline are slowed this would promote a stronger active state, and be the opposite of the reduced EF and FS.

Fourth, while this leaves the door open for a potential myofilament associated defect in contractility caused by the PKB KO (Fig 1) or SPEG-3A mice, that notion is not addressed at all (and would be opposed to their central conclusion).

Fifth, there are no measurements at all of either SR Ca content or SR Ca leak. Without that sort of data, it is hard to interpret what is really going on with myocyte Ca handling. For example, if reduced SR Ca uptake is responsible for the slowed [Ca]_i decline in PKB-KO (Fig 1), SPEG-3A KI (Fig 6) and SPEG D3098A (Fig 4), then that ought to reduce SR Ca content and reduce Ca transient amplitude. But in the one case where Ca transient amplitudes are reported, they are unchanged. That needs mechanistic explanation. Ca spark decline rate is primarily a function of diffusion and RyR properties and SERCA2 function is a minor contributor. Thus, without data on SR Ca load or leak (via RyR), the spark data is also difficult to assign mechanistic interpretation. Sixth, phospholamban phosphorylation is the main acute regulator of SERCA2a function in adult ventricular myocytes, and this seems essentially unconsidered or explored.

- The amplitudes of Ca²⁺ transients in the PKBα/β KO mice displayed no change as compared to those in the control mice at 4 weeks after tamoxifen induction (the new Fig. 1F). The amplitudes of Ca²⁺ transients were not altered in the Speg^{3A} knockin mice at the age of 3 months (the new Fig. 8J), but significantly decreased at the age of 7 months (the new Supple. Fig. 9A). This phenomenon is reminiscent of changes of Ca²⁺ transients in SPEG KO hearts in which increased FDHM and Tau also precede decreased amplitudes of Ca²⁺ transients (PMID: 30566039). We also studied Ca²⁺ transients in primary neonatal rat cardiomyocytes overexpressing the SPEG WT and 3A mutant proteins. The FDHM

and Tau of Ca^{2+} transients were larger in cells expressing the SPEG^{3A} mutant protein than those in cells expressing the SPEG WT protein (new Fig. 4K). Importantly, the amplitudes of Ca^{2+} transients were significantly lower in cells expressing the SPEG^{3A} mutant protein than those in cells expressing the SPEG WT protein (new Fig. 4K). To further address the effect of SPEG^{3A} knockin mutation on SERCA2a, we isolated the microsomes and determined the ATPase activity and Ca^{2+} transport activity of SERCA2a. We found that the ATPase activity of SERCA2a was normal but its Ca^{2+} transport activity was significantly decreased in the SPEG^{3A} knockin mice (new Fig. 8G-H). We determined phosphorylation of phospholamban in the heart of SPEG^{3A} knockin mice, and found it was not altered (new Supple. Fig. 8B-C). Phosphorylation of RyR2 was also not changed in these hearts (new Supple. Fig. 8B-C), which was in agreement with unaltered frequency of Ca^{2+} sparks in SPEG^{3A} knockin cardiomyocytes (new Fig. 8K). These data suggest that SR Ca^{2+} leak is most likely unaltered in the SPEG^{3A} knockin hearts. As suggested, we determined expression of myofilaments in the heart of PKB α / β KO and SPEG^{3A} knockin mice. Expression of tropomyosin-3 (TPM3) and cardiac troponin I (cTnI) were normal in the heart of both PKB α / β KO and SPEG^{3A} knockin mice (new Supple. Fig. 3A, 3C, 8B, and 8D).

2. The SR Ca-ATPase is known to be a energetically highly efficient, with nearly 2 Ca transported for each ATP used, and able to create a [Ca] gradient across the SR membrane that approaches the thermodynamic limit. Claiming a major increase in Ca transport efficiency (more Ca transported per ATP) is an issue. Moreover, their prior study implied coupling ratios above 100 Ca transported per ATP split (which thermodynamically is impossible). This gives one concern as to the precise mechanism by which phosphorylation of SERCA2a at T484 enhances SERCA2 function. This was not especially well addressed in the prior study or here.

- We sincerely thank the reviewer for pointing out the issue of Ca^{2+} transport efficiency, which helps us to identify some typos that were inadvertently made in our previous study. The unit for Ca^{2+} transport should be pmol/sec/mg instead of nM/sec/mg, and the unit for ATPase activity should be nmol/min/mg instead of nM/min/mg in our previous study. We deeply apologize for the inconvenience these typos may cause to the reviewer and the readers of the journal. We have already made corrigendum for these typos (PMID: 31804910).

In the context of these corrected units, our previous data indicated 1/10 to 1/5 of Ca^{2+} transported per ATP split. This low Ca^{2+} transport efficiency might be due to the fact that Ca^{2+} transport and ATPase activity of SERCA2a were measured separately using totally different assay systems. The assay systems for these two activities are very robust. We determined Ca^{2+} transport and ATPase activity using microsomes from the Speg^{3A} knockin hearts, and found that they are in similar ranges as our previous study. In agreement with decreased SERCA2a phosphorylation, the SR Ca^{2+} transport was decreased while the ATPase activity of SERCA2a was not altered in the Speg^{3A} knockin hearts (new Fig. 8G-H).

We agree that it is important to elucidate the precise mechanism by which phosphorylation of SERCA2a at T484 enhances SERCA2 function. This may not be an easy question to answer, and various approaches need to be applied to solve this critical question, including biochemistry, cell biology, and structure biology. We recently took a genetic approach to examine the importance of SERCA2a-Thr⁴⁸⁴ phosphorylation *in vivo*. We generated a SERCA2a^{T484A} knockin mouse in which Thr⁴⁸⁴ on SERCA2a was mutated to a non-phosphorylatable alanine. Our preliminary data show that cardiac function was impaired in the SERCA2a^{T484A} knockin mice, demonstrating the importance of SERCA2a-Thr⁴⁸⁴ phosphorylation *in vivo* (Letter Fig. 1A-C). We will utilize this model in

combination with other approaches to investigate the mechanism by which phosphorylation of SERCA2a at T484 enhances SERCA2 function.

3. PKB has also been shown to enhance SR Ca leak via CaMKII-dependent phosphorylation of RyR2 (PMID: 28476660). Conceivably there is more than one thing going on, and increased SR Ca leak combined with enhanced SR Ca uptake could result in unaltered SR Ca load and Ca transient amplitudes. That is not probed.

- As the reviewer pointed out, PKB mediates β -adrenergic receptor (β -AR) induced SR Ca leak via CaMKII-dependent phosphorylation of RyR2 (PMID: 28476660). Direct activation of Epac by 8-CPT (8-(4-chlorophenylthio)-2'-O-methyl-cAMP) mimicked β -AR-induced SR Ca²⁺ leak (PMID: 28476660). A PI3K inhibitor LY294002 prevented 8-CPT-induced Ca²⁺ spark frequency (CaSpF) but not baseline CaSpF (PMID: 28476660), suggesting that PKB may mediate RyR2 phosphorylation under hypertrophic conditions but probably not under basal conditions. In agreement with this notion, we determined RyR2 phosphorylation, and found no alteration in PKB α / β KO hearts (new Supple. Fig. 3A-B). We also include a discussion in the revised manuscript (Page 8 line 24-28).

4. There are numerous cases where controls are missing entirely (e.g. why isn't PKB-KO shown at baseline without insulin in Fig 2C). There are several other examples of this sort of oversight. It is stated that PKB deficiency causes insulin resistance in mouse (pg 5), but that is neither shown nor referenced.

- In the revised manuscript, we re-analyzed the PKB α / β KO hearts stimulated with or without insulin including proper controls, and showed that insulin-induced phosphorylation of PKB substrates AS160 and GSK3 was blunted in these hearts (new Supple. Fig. 1D). We also isolated PKB α / β KO cardiomyocytes, and examined phosphorylation of SPEG and SERCA2a. Insulin-stimulated PAS-reactive phosphorylation of SPEG was decreased in these cells (new Fig. 5A-B in the revised manuscript). Furthermore, we found insulin increased SERCA2a-Thr⁴⁸⁴ phosphorylation in the WT cardiomyocytes (new Fig. 5A-B in the revised manuscript). In contrast, insulin did not increase SERCA2a-Thr⁴⁸⁴ phosphorylation in the PKB α / β KO cardiomyocytes (new Fig. 5A-B in the revised manuscript). These data show that insulin resistance occurs in the PKB α / β KO hearts.

5. The link of this PKB-SPEG-SERCA pathway in diabetic cardiomyopathy is lost in the shuffle. In many overexpression studies in HEK cells (and even in myocytes) it is unclear that diabetes or insulin activation are involved in the measurements (e.g. it does not look like insulin was required to activate PKB/SPEG effects in Fig 4). Also many gels are neither quantified or statistically analyzed (e.g. Figs 2-4).

- As above-mentioned, we analyzed the SPEG-SERCA2a pathway in the PKB α / β KO hearts in response to insulin, and found that the insulin responses such as PAS-reactive phosphorylation of SPEG and SERCA2a-Thr484 phosphorylation were inhibited in these hearts (new Fig. 5A-B in the revised manuscript). Furthermore, we examined the PKB-SPEG-SERCA2a pathway in the hearts of diet-induced obese mice, and found that the insulin-induced phosphorylation of PKB, SPEG and SERCA2a was inhibited in these hearts (new Fig. 6D-E in the revised manuscript). These data highlight the impairment of PKB-SPEG-SERCA2a pathway in insulin resistant heart. As suggested, we quantified

western blots as many as we can throughout the manuscript, and performed statistical analysis.

Reviewer #3 (Remarks to the Author):

This is an interesting manuscript by Quan and colleagues potentially linking PKB-SPEG signaling with insulin resistance and the subsequent consequences of abnormal calcium homeostasis in cardiomyocytes during diabetic cardiomyopathy. The experiments were elegantly performed, and the results are convincing using genetically modified mouse models of disease. Insulin resistance in the mouse heart was shown using an inducible PKB α/β deficient mouse. Moreover, a Speg3A knockin mutant mouse supported the in vivo function of Speg phosphorylation by insulin.

Mutation of the Speg gene has been shown previously to lead to a dilated cardiomyopathy in the developing heart and death shortly after birth. Speg mutations have also been associated with a dilated cardiomyopathy in human patients. Moreover, inducible disruption of Speg in the heart has shown altered calcium homeostasis and cardiac dysfunction. The present manuscript advances our understanding and provides new insight into the physiologic function of Speg, with a focus on the abnormal consequences of diabetic cardiomyopathy and abnormal calcium homeostasis.

- We thank this reviewer for finding our manuscript 'interesting', and highly appreciate his/her comments 'the experiments were elegantly performed, and the results are convincing'.

While the authors make interesting points about the ability to promote insulin resistance in the hearts without other metabolic changes, I believe it is still critical to demonstrate that in a rodent model of diabetes or obesity-induced insulin resistance (and not a genetically modified mouse) that there is evidence of altered Speg phosphorylation in the proposed serine/threonine residues. If this does occur, is there altered calcium homeostasis (including decreased phosphorylation of Thr484 on SERCA2a) and associated cardiomyopathy. Using a pathophysiologic model of diabetes or obesity-induced insulin resistance, even in the presence of metabolic abnormalities, will provide additional support to the underlying conclusion of the manuscript regarding Speg, insulin resistance, and diabetic cardiomyopathy.

- As suggested, we utilized hearts from DIO mice as a model of diabetic cardiomyopathy. As expected, cardiac function was impaired in DIO mice (new Fig. 6). We examined the PKB-SPEG-SERCA2a pathway in the hearts of DIO mice, and found that the insulin-induced phosphorylation of PKB, SPEG and SERCA2a was inhibited in these hearts (new Fig. 6 in the revised manuscript). Moreover, the FDHM and Tau of calcium transients were increased and their peaks were decreased in the cardiomyocytes of DIO mice (new Fig. 6).

The description of the statistical analyses appears to be appropriate, however providing further information about the specific statistical analysis performed for each figure in the legend would help the readers to better understand the analysis.

- Thanks for pointing this out, and we include the information on the specific statistical analysis in the legend of each figure in the revised manuscript.

Reviewer #4 (Remarks to the Author):

Report on manuscript submitted to Nature Communications titled:
A PKB-SPEG signaling nexus links insulin resistance with diabetic cardiomyopathy by regulating calcium homeostasis"

Authors: Quan C et al from the research group of Prof S Chen

The authors delineated the signalling events leading from the activation of PKB/Akt to SERCA2a phosphorylation and the function of re-uptake of calcium by the SR. To do so, they utilized biochemical techniques and a mouse model with a specific knockin mutation of the protein of interest, SPEG. This research was built on their previous work showing that SPEG controls calcium re-uptake into the sarcoplasmic reticulum.

The investigation was well planned and formed a cohesive set of evidence to prove the interactions as described. The research generated an enormous amount of very interesting results. The evidence supplied in the paper substantiates the conclusion. The findings are novel and will be of interest to a wider spectrum of researchers, both from the field of insulin resistance as well as cardiac contractility.

- We thank this reviewer for the supportive comments, and highly appreciate that he considers 'the findings are novel and will be of interest to a wider spectrum of researchers, both from the field of insulin resistance as well as cardiac contractility'.

I do have however, have a broad comment that the model utilized simulated a metabolic defect that will arise in a state of insulin resistance – as also stated in the rationale of the research project. Despite the fact that the impairments were observed without metabolic derangements in the animals, to conclude that "a diabetic cardiomyopathy" can develop independent of metabolic insults, is not completely correct. Can the authors list any other insult that will allow the SPEG protein to not be phosphorylated by PKB? If not, they have actually demonstrated an end-point of the result of metabolic derangements and should change their conclusion as such.

- We agree with the reviewer, and have changed the conclusion as 'cardiac insulin resistance impairs calcium homeostasis via the PKB-SPEG-SERCA2a pathway, which contributes to the development of diabetic cardiomyopathy' in the revised manuscript. We delete 'independent of metabolic insults' throughout the manuscript.

It is stated in the methods that different statistical analyses were performed on data sets but it should be listed in the Figure legends, together with the n-values, what analysis method was performed to analyse that specific set of results.

- As suggested, we include the information on the specific statistical analysis together with the n-values in the legend of each figure in the revised manuscript.

Minor points:

On page 10, second paragraph line 5: remove one of the "in" word.

- Changed as suggested (Page 12 line 16)

Page 35, supplementary table 2: In the legend it is stated that the work was performed using IGF1 as ligand but the table states Insulin + or -. Please correct.

- Corrected as suggested (new Supple. Table 2)

Reference 16 is incomplete

- Corrected as suggested (new Reference 16)

[Redacted]

REVIEWERS' COMMENTS:

Reviewer #1 (Remarks to the Author):

The authors adequately answered to all my comments. The additional data provided greatly enhanced the impact of the manuscript

Reviewer #2 (Remarks to the Author):

This has been a highly responsive revision of the previously reviewed manuscript. They have largely addressed the criticisms that I raised in my prior critique. The paper is strengthened and the conclusions are more compelling, based in part on much additional data that has been added to the paper.

Reviewer #3 (Remarks to the Author):

The authors have addressed my concerns in the revised manuscript.

Reviewer #4 (Remarks to the Author):

The authors extensively revamped the paper and added additional results to strengthen the work. They have adequately addressed all the queries I initially raised against the paper and corrected all the mistakes. They also added the relevant statistical analyses to the individual figures to clarify that I advised instead of just a general umbrella statement.

i have no further comments.

Ref: Ms. NCOMMS-19-07467A

Title: A PKB-SPEG signaling nexus links insulin resistance with diabetic cardiomyopathy by regulating calcium homeostasis

Thanks a lot to the reviewers for the positive and helpful reviews of our paper. Here we outline our responses to comments from the reviewers.

REVIEWERS' COMMENTS:

Reviewer #1 (Remarks to the Author):

The authors adequately answered to all my comments. The additional data provided greatly enhanced the impact of the manuscript

- We thank this reviewer for the critical review of our manuscript, which helps to improve our study.

Reviewer #2 (Remarks to the Author):

This has been a highly responsive revision of the previously reviewed manuscript. They have largely addressed the criticisms that I raised in my prior critique. The paper is strengthened and the conclusions are more compelling, based in part on much additional data that has been added to the paper.

- We thank this reviewer for the constructive review of our manuscript, which helps to strengthen our conclusions.

Reviewer #3 (Remarks to the Author):

The authors have addressed my concerns in the revised manuscript.

- We thank this reviewer for the helpful review of our manuscript, which helps to strengthen our work.

Reviewer #4 (Remarks to the Author):

The authors extensively revamped the paper and added additional results to strengthen the work. They have adequately addressed all the queries I initially raised against the paper and corrected all the mistakes. They also added the relevant statistical analyses to the individual figures to clarify that I advised

instead of just a general umbrella statement. i
have no further comments.

- We thank this reviewer for the constructive review of our manuscript, which
helps to improve our work.